# iCLAP: an innovative method for integrable co-detection of low-abundance antigens with high-plex immunostaining

Fan Wu [1,2], Shuyuan Zheng[1], Yani Chen[1], Peijia Ye[3], Moo Joong Kim[1], Seojin Lee[1], Geroge Kuo[4], Shriya Pillan[1], Ruihan Yuan [3], Kyu Sang Han[1], Bofei Yu[3], Qingfeng Zhu [5,6], Sarah M. Shin [5], Courtney D. Cannon[5], Gabriele Pierre[1], Kanako Iwasaki [7], Cristina Aguayo-Mazzucato [7], Nicolas Musi[8], George A. Kuchel [9], Birgit Schilling [10], Laura D. Wood[11], Won Jin Ho [5], Robert A. Anders[5,6], Denis Wirtz [1,2,3,5,11,12] ✉ & Pei-Hsun Wu [1,2] ✉

Multiplexed protein imaging enables spatial analysis of complex tissues, but detecting proteins expressed at low levels remains challenging, particularly in widely available formalin-fixed, paraffin-embedded (FFPE) specimens. Many biologically important regulators—including senescence markers, transcription factors, and secreted proteins—are therefore difficult to study in situ using existing high-plex methods. Here we show that integrable Co-detection of Low-Abundant Proteins (iCLAP) enables sensitive and highly multiplexed protein detection within the same FFPE tissue section. iCLAP combines iterative signal amplification with efficient fluorophore inactivation, enabling repeated staining of the same tissue section and seamless integration with established multiplex imaging platforms to achieve profiling of more than 40 markers. Application of iCLAP to human pancreatic tissues revealed spatially distinct senescence-associated protein patterns across tissue compartments. Together, iCLAP expands the analytical capabilities of FFPE tissues, enabling high-sensitivity, high-dimensional spatial proteomic studies of complex biological processes.

In-depth tissue profiling with spatially resolved information is essential for uncovering pathological processes at the molecular, cellular, and tissue levels in situ[1–3]. Understanding dysregulated molecular and cellular processes is crucial for identifying prognostic markers and therapeutic targets in human diseases. However, tissues exhibit highly complex landscapes with intricate spatial arrangements of diverse cells and tissue types, making them challenging to study. Immuno-based detection methods have been critical for visualizing antigen and protein locations in archived human tissue specimens. High-plex tools such as co-detection by indexing

[1]Department of Chemical and Biomolecular Engineering, The Johns Hopkins University, Baltimore, Maryland, USA. [2]Institute for NanoBioTechnology, The Johns Hopkins University, Baltimore, Maryland, USA. [3]Department of Biomedical Engineering, The Johns Hopkins University, Baltimore, Maryland, USA. [4]Department of Molecular and Cellular Biology, The Johns Hopkins University,, Baltimore, Maryland, USA. [5]Department of Oncology, The Johns Hopkins University School of Medicine, Baltimore, Maryland, USA. [6]Convergence Institute, The Johns Hopkins University School of Medicine, Baltimore, Maryland, USA. [7]Joslin Diabetes Center, Harvard Medical School, Boston, Massachusetts, USA. [8]Department of Medicine, Cedars-Sinai Medical Center, Los Angeles, California, USA. [9]UConn Center on Aging, University of Connecticut, Farmington, Connecticut, USA. [10]Buck Institute for Research on Aging, Novato, California, USA. [11]Department of Pathology, The Johns Hopkins University School of Medicine, Baltimore, Maryland, USA. [12]Johns Hopkins Physical Sciences - Oncology Center, The Johns Hopkins University, Baltimore, Maryland, USA. ✉e-mail: wirtz@jhu.edu; pwu@jhu.edu

(CODEX)[4], cyclic immunofluorescence (CyCIF)[5], imaging mass cytometry (IMC)[6], and 4i[7] enable simultaneous detection of key markers that define tissue compartments and cell types, allowing for a comprehensive mapping of tissue architecture and disease pathology. Due to their intrinsic detection limits, these approaches typically focus on highly expressed proteins that are uniquely associated with specific cell or tissue types.

However, many critical biological processes—such as cancer stemness[8], cellular senescence[9], quiescence[10], immune evasion[11], transcriptional regulation[12], and neurotransmitter signaling[13]—are characterized by the differential expression of low-abundance proteins. For example, well-known immune evasion markers and cancer prognostic factors PD-1, LAG-3, and CTLA-4 are typically in low abundance[14]. Detecting these proteins with sufficient sensitivity remains a major challenge[15]. Since high-plex imaging platforms rely on primary conjugated antibodies, their sensitivity is inherently limited to detect low-abundance proteins[16–18].

Cellular senescence plays a crucial role in aging and age-associated diseases, as well as in conditions such as cancer[19–22]. Extensive studies in both in vitro and in vivo models have identified key senescence-associated proteins, including P16, P21, P53, 53BP1, HMGB1, and LaminB1[22–28]. However, many of these proteins are difficult to detect in archival tissue specimens, and how their collective expression defines cellular senescence remains poorly understood. Co-detecting multiple senescence markers alongside tissue and cell-type markers in human tissue is essential for elucidating the role of cellular senescence in situ. Highly sensitive protein detection strategies, such as tyramide signal amplification (TSA), enhance

fluorescence signal intensity via enzymatic deposition of tyramide molecules[29]. While effective, TSA-based approaches typically allow for the detection of no more than eight markers per section[30–35], limiting their ability to capture the full complexity of senescence in tissues.

Here, we introduce iCLAP (Integrable Co-detection of Low-Abundance Proteins) – an innovative platform to integrate TSA-based signal amplification into high-dimensional spatial proteomics workflows such as IMC, CyCIF, and CODEX. The core innovation of iCLAP lies in its optimized bleaching technology, which completely removes TSA-conjugated signals between staining cycles without degrading tissue morphology or antigenicity. This approach enables iterative TSA applications, a capability previously unattainable in highly multiplexed platforms. By overcoming the limitations of traditional high-plex imaging, iCLAP enables high-sensitivity detection of low-abundance proteins while preserving the ability to map complex tissue landscapes, providing a powerful tool for studying disease pathology with enhanced spatial and molecular resolution.

## Results
### iCLAP multiplex staining workflow
In iCLAP, low-abundance proteins and antigens were labeled using cyclic TSA-based staining. To enable sequential rounds of staining, we developed and optimized a fluorophore inactivation method to effectively remove TSA-labeled signals from tissue sections. Once the fluorophores were inactivated, the same tissue section could undergo additional rounds of cyclic TSA staining or be combined with other multiplex immunostaining methods, such as IF, CyCIF, CODEX, or IMC, enabling high-plex detection within the same tissue section (Fig. 1a).

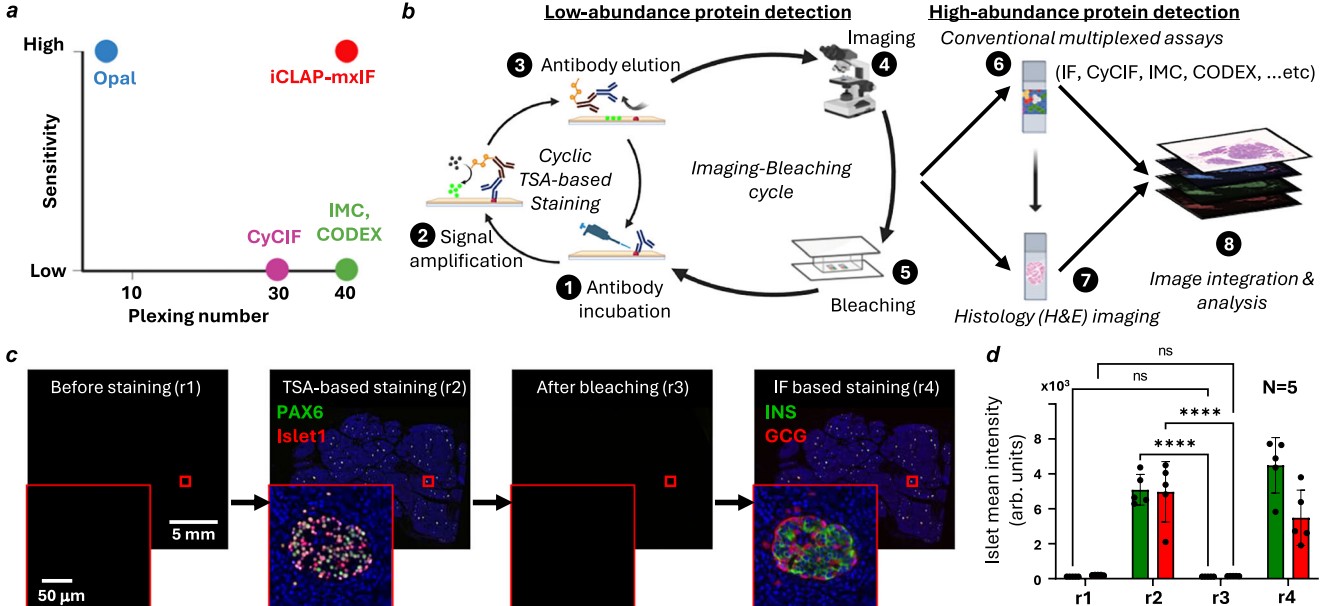

**Fig. 1 | iCLAP workflow achieves sensitive and high-plex protein detection in FFPE tissue via cyclic TSA amplification and efficient bleaching. a** Sensitivity and plexing capacity comparison of iCLAP-mxIF with other existing multiplex methods. **b** Schematic of the integrated iCLAP workflow. Low-abundance proteins are stained with cyclic tyramide-based signal amplification (TSA) method, and high abundance proteins are then stained with conventional multiplexing immunolabeling assays such as IF, CyCIF, IMC, ...etc. A novel fluorophore inactivation method is applied between imaging cycles to efficiently reduce the fluorescent signal in the previous round to a background level. Created with BioRender.com. Wu, F. (2025) https://BioRender.com/vngrd8o. **c** Representative whole slide background images were obtained before iCLAP staining (imaging round 1, r1). Representative whole slide iCLAP images of a human pancreatic tissue first stained for Islets of Langerhans developmental transcription factors PAX6 (green) and Islet1 (red) with TSA method (imaging round 2, r2) and then bleached with novel fluorescence removal method

(imaging round 3, r3), followed by insulin (green) and glucagon (red) staining with IF (imaging round 4, r4). **d** Five islets of Langerhans regions were selected for quantitative fluorescence intensity analysis across different stages of staining cycles. Analysis results show the novel bleaching method significantly reduced the fluorescence intensity (in arbitrary units) in stained channels and reduced the fluorescence intensity to the background level. Green (Cy5) and red (TRITC) bars correspond to different microscope imaging channels. Statistical analysis was performed using two-way ANOVA (two-sided) followed by Tukey's multiple comparisons test. No significant differences were observed between r1 and r3 for either channel (r1 green vs r3 green, $P = 1.00000000$; r1 red vs r3 red, $P = 0.999999997$). In contrast, fluorescence intensity differed significantly between r2 and r3 for both green ($P = 4.12 \times 10^{-7}$) and red ($P = 7.66 \times 10^{-7}$). Data are presented as mean ± SD from five biological replicates.

Chromogenic histological staining, such as Hematoxylin and Eosin (H&E), could also be performed on the same section following the iCLAP procedure (Fig. 1b). Here, we demonstrate that the intense fluorescence signals of two pan-islet markers, PAX6 and Islet1, from TSA-based labeling could be reduced to near-background levels using the proposed iCLAP method in a human pancreatic tissue (Fig. 1c, d). Furthermore, we showed that the same tissue section could be subsequently stained with fluorophore-conjugated antibodies targeting other islet markers, including insulin and glucagon. All four islet markers were co-localized in the islet region of the pancreas, highlighting the effectiveness of the iCLAP workflow (Fig. 1c, d). We note that our fluorophore inactivation procedure is more effective at removing the strong fluorophore signal from TSA-based staining compared to the CyCIF protocol[5] (Supplementary Fig. 1a, b). We also quantitatively evaluated tissue section integrity across staining cycles in both iCLAP and CyCIF. We found that iCLAP resulted in approximately 2% tissue loss after 10 staining cycles, comparable to the CyCIF procedure in our tests (Supplementary Fig. 2) and consistent with previously reported levels of tissue loss in CyCIF studies[36].

## Signal amplification enables highly sensitive detection of key senescence markers

Many senescence markers are difficult to detect in archival tissues due to their low levels of expression[24,37,38]. We compared staining protocols for senescence markers P16, P21, P53 and 53BP1, using immunofluorescence with TSA and with traditional IF procedure (i.e. primary antibodies + secondary antibody with a conjugated fluorophore) in archival FFPE tissue sections from human pancreas (Fig. 2). Based on TSA-based immunofluorescence, we observed that P16 exhibited a predominantly cytosolic staining pattern, while P53, P21, and 53BP1 primarily displayed nuclear staining (Fig. 2). Nuclei with higher P53 and P21 signals were frequently observed in the exocrine tissues of the pancreas (i.e., acini), whereas nuclei with strong 53BP1 signal typically localized within endocrine tissue compartments (i.e., islets) of the pancreas (Fig. 2). Consistent with previous reports, we found that the clustering of P16-positive cells was primarily located in the pancreatic islets[39,40]. The TSA-based immunofluorescence staining of P53, P16, and P21 were further validated with IHC staining using antibodies from multiple sources targeting the same antigen (Supplementary Fig. 3a, b).

In contrast to TSA amplification-based immunofluorescence, IF staining of P16, P21, P53 and 53BP1 with the same antibody dilution/ incubation time was weak, leading to a nearly undetectable fluorescence signal (Fig. 2). For P16 and 53BP1, we found that with prolonged incubation times and/or high concentrations of antibodies, the stained signal became more apparent, but with much lower intensity than TSA-based immunofluorescence. For example, we found that the staining intensity for 53BP1 signal improved more than 10-fold with the traditional IF method when using a 20-fold higher concentration of antibody and 20-fold longer incubation time (Fig. 2a, b). For P16 staining, the prolonged incubation (12 h vs. 40 min) of the tested antibody, which was pre-diluted and ready to use, improved the staining signal (Fig. 2c, f, g). Nevertheless, quantitative analysis showed that the signal-to-noise ratio (SNR) for positively stained regions (e.g., islets) in P16 was 10-fold lower for traditional IF compared to TSA, even with prolonged antibody incubation time. For P53 and P21, clear positive staining signals were not detected using traditional IF staining, even after extending antibody incubation times and increasing antibody concentrations (Fig. 2d, e, Supplementary Fig. 4a, b).

We tested a total of 29 antibodies targeting different senescence markers in FFPE tissues and found that their respective antigens, including senescence-associated secretory factors, phosphorylated proteins, and transcription factors, were effectively stained using TSA-based immunofluorescence (Supplementary Fig. 5a–f, Supplementary Data 2). Notably, HMGB1 and Lamin B1 exhibited adequate staining

with traditional immunofluorescence. Together, these results underline the importance of TSA amplification for the sensitive detection of specific cellular senescence markers in archival FFPE tissues.

## iCLAP-IF enables 6-plex detection of senescence markers in healthy pancreas

By integrating the iCLAP workflow with immunofluorescence (iCLAP-IF), we established an antibody panel for co-detecting six conventional senescence markers—P53, P21, P16, 53BP1, HMGB1, and Lamin B1—in archival FFPE tissues. In this iCLAP-IF approach, low-abundance markers P53, P21, P16, and 53BP1 were detected using TSA-based immunofluorescence, while Lamin B1 and HMGB1 were detected with fluorophore-conjugated antibodies (Fig. 3a). Detection and imaging occurred in two rounds: first, P53, P21, and P16 were detected, followed by staining and imaging of 53BP1, HMGB1, and Lamin B1 after fluorophore signal inactivation from the first round.

iCLAP-IF enhances workflow efficiency for multiplex staining, as TSA-based antigen detection is more time-consuming than the IF process. We profiled the fluorescence intensity of these senescence markers in the healthy pancreas of a 63-year-old male using the 6-plex senescence panel (Fig. 3b). The staining quality obtained with iCLAP-IF was comparable to that of individually stained controls, showing similar peak intensity and distribution profiles (Supplementary Fig. 6a–f), thereby validating the robustness of the iCLAP multiplexing workflow. The co-detection of these markers allows us to analyze co-expression patterns of senescence markers at the single-cell level in healthy pancreatic parenchyma.

We found that in acinar regions, cells with relatively high P21 and P53 signals arose from different cells (Fig. 3c, Supplementary Fig. 7a, b). We further employed clustering analysis based on senescence marker intensities from more than 70,000 individual cells and identified 9 subtypes in healthy pancreatic parenchyma (Fig. 3d). Fluorescence intensity of markers was used to distinguish high- and low-expression cell populations on the same slide. We found that a substantial fraction of cells exhibited relatively low expression across all probed markers, corresponding to cluster 1, which accounted for approximately 30% of all cells. Distinct clusters displayed enrichment for single markers, including P53 (cluster 4), P21 (cluster 6), and P16 (cluster 9), whereas notable populations with dual or triple P53/P21/ P16 expression were not observed (Fig. 3e). These marker-enriched clusters were further characterized by elevated levels of 53BP1 and low levels of LaminB1 and HMGB1.

We explored the fluorescence intensity landscape of the stained markers at the single-cell level using UMAP. Integrating the clustering analysis results on the UMAP landscape revealed four distinct niches emerging from cells with low expression (Fig. 3f). These niches were associated with high levels of P16/53BP1 (cluster 9), P21/53BP1 (cluster 6), P53 (cluster 4), and DAPI/HMGB1(cluster 8), respectively. Together, these results suggest that senescence markers are generally exclusively expressed at the single-cell level and that different markers are likely linked to distinct senescence processes/pathways in situ. Importantly, both the staining patterns and the UMAP embedding, including concordant cluster frequencies, were consistently reproduced in adjacent sections (Supplementary Fig. 7c–f).

We further explored the spatial distribution of senescence marker-defined subtypes in pancreatic tissues. We found that senescence marker intensity was highly associated with specific tissue compartments in the pancreas (Supplementary Fig. 8a–d). Notably, subtypes with relatively high P16 and 53BP1 expression (cluster 9) were predominantly localized to the islets (Fig. 3g), whereas P53-positive cells (cluster 4) were enriched in the acinar compartment. In epithelial (ductal) regions, we found that a substantial population of cells were associated with positive P21 (clusters 6). We further applied the 6-plex iCLAP-IF panel to pancreatic tissue from four additional human donors (young, n = 2; old, n = 2) and observed consistent staining quality and

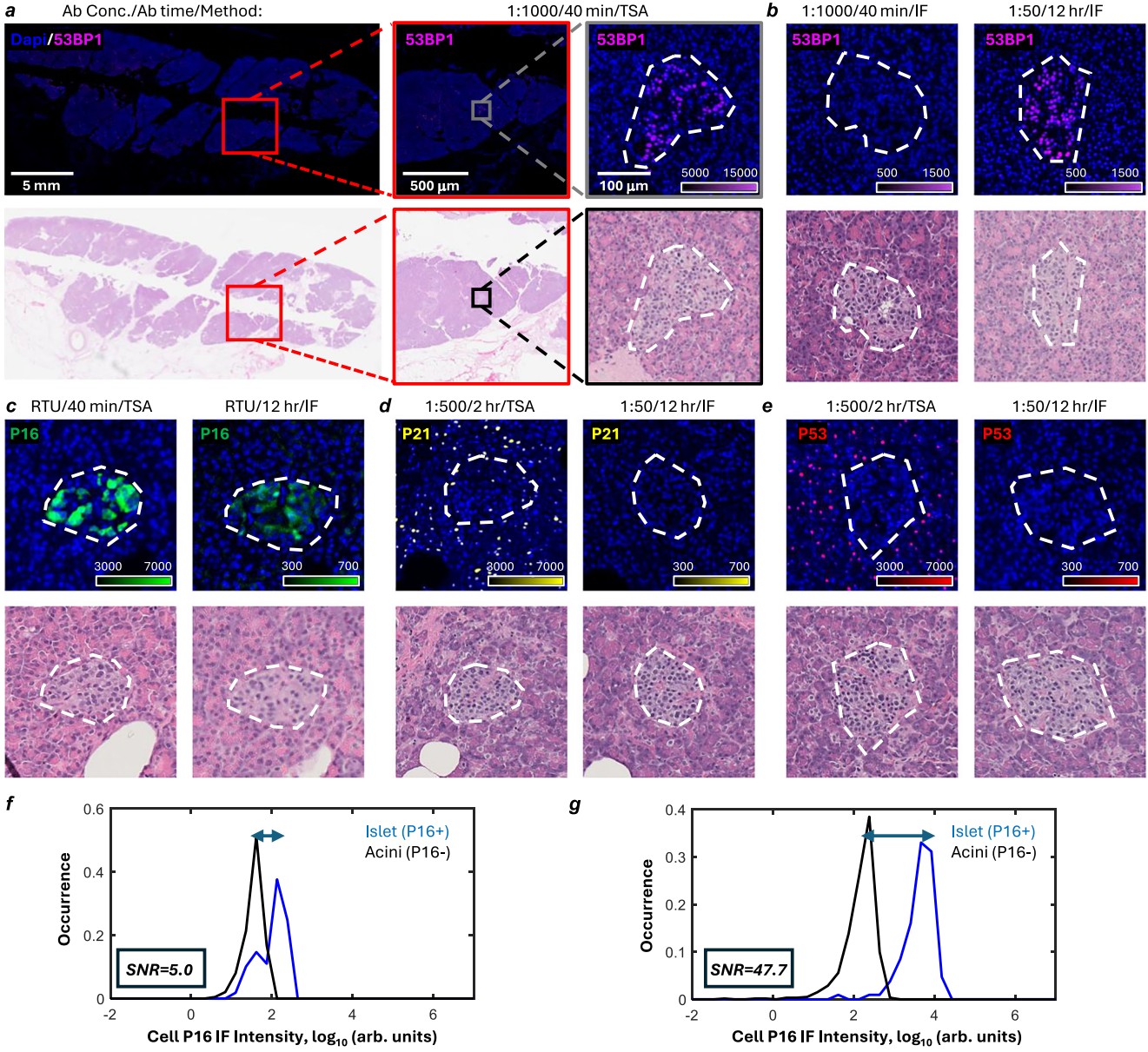

**Fig. 2 | Signal amplification enables sensitive detection of low abundance senescence markers in healthy pancreas. a–c** Images showing 53BP1 staining on human pancreatic tissue using TSA-based immunofluorescence (TSA) with a 1:1000 diluted antibody and 40-min incubation (**a**), indirect immunofluorescence (IF) with a 1:1000 diluted antibody and 40 min incubation (**b**), and indirect immunofluorescence with a 1:50 diluted antibody and 12 h incubation (**c**). Corresponding H&E images from adjacent sections of the same field are also included. **d** P21 staining of 2 h incubation of 1:1000 diluted antibody and 12 h with 1:50 diluted antibody with TSA and IF, respectively. **e** P53 staining (red) of 2 h incubation of 1:1000 diluted antibody and 12 h with 1:50 diluted antibody with TSA and IF, respectively. **f, g** Histograms show the distribution of cell P16 intensity in islet and acini regions using indirect immunofluorescence staining (**f**) or TSA-based immunofluorescent (**g**). All immunofluorescence images shown are representative of independent staining experiments performed on three independent human pancreatic tissue samples (SENPAN_1, SENPAN_2, and SENPAN_3; see Supplementary Data 1), with similar results observed. The pseudocolor intensity scale bar shown at the bottom right indicates fluorescence intensity values (arbitrary units).

senescence marker–associated patterns across donors (Supplementary Fig. 9a). At the islet level, the frequency of P16[+] cells was higher in older compared to younger donors, whereas the frequency of 53BP1[+] cells did not differ by age (Supplementary Fig. 9b–e). These results demonstrate that the compartment-associated senescence signatures identified by iCLAP-IF are reproducible across biological replicates and reflect expected age-related patterns[41]. Together, our iCLAP analysis highlights the complex landscape of senescence marker signatures at the single-cell level and its strong association with specific tissue compartments, emphasizing the importance of multiplex tools in uncovering spatial cellular senescence in situ.

We explored whether iCLAP-based multiplex senescence detection could be extended beyond pancreatic tissue. We applied iCLAP-based 6-plex senescence marker detection to cervical, breast, ovarian, skin, and liver tissues from both normal and tumour samples (Supplementary Fig. 10a, b). Our results showed that senescence markers P21, P53, and P16 were substantially expressed in tumour tissues across all tested tissue types. In contrast, healthy tissues exhibited low levels of senescence marker intensity signal. Overall, these results demonstrate that iCLAP is an effective and robust method to measure the co-variation in the intensity profiles among multiple senescence markers in FFPE tissues.

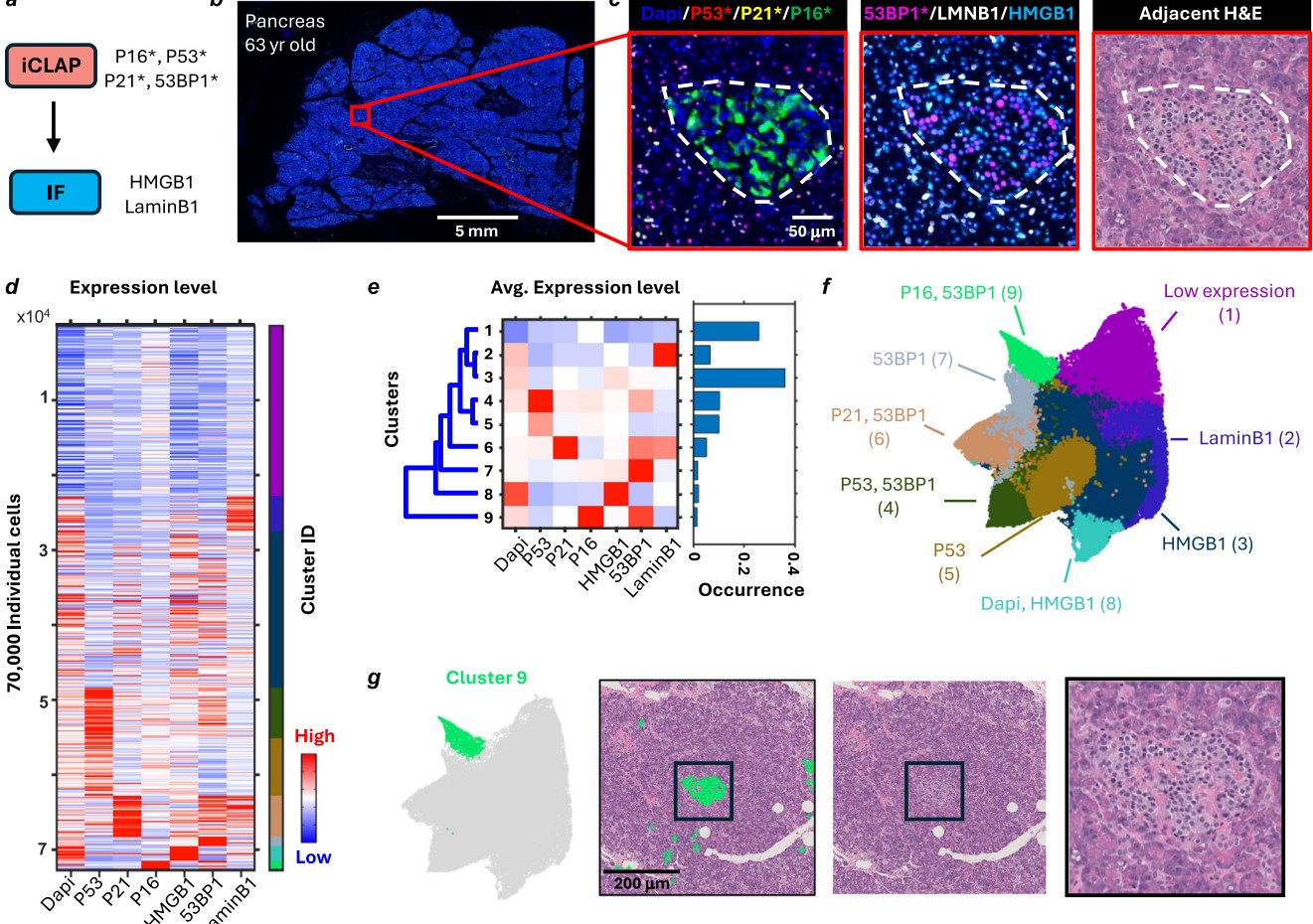

**Fig. 3 | Multiplex senescence marker detection with iCLAP-IF reveals sub-populations of cells with distinct marker expression patterns in a human pancreatic tissue. a** Schematic of the 6-plex iCLAP-IF workflow for senescence marker detection. P16, P53, and P21 were stained and imaged in the first round, while 53BP1, HMGB1, and Lamin B1 were detected in the second round. TSA was used for P16, P53, P21, and 53BP1 detection, whereas HMGB1 and Lamin B1 were visualized using conventional immunofluorescence. **b** Whole slide image of a healthy pancreatic tissue section from a 63-year-old donor stained with 6-plex senescence markers. **c** Zoom-in fluorescence and H&E images of a representative islet of the Langerhans region. The observed patterns were reproduced in 5 adjacent independent repeats within the same tissue sample (SENPAN_2) and across three independent human pancreatic tissue samples (SENPAN_1, SENPAN_2, and SENPAN_4; see Supplementary Data 1). **d** A heatmap displaying K-means clustering results based on stained senescence marker expression in over 70,000 cells of the pancreatic parenchyma selected from ten non-overlapping ROIs (2000 × 2000 px; ~1.3 × 1.3 mm each). **e** A heatmap shows the intensity profiles of 9 clusters. The bar graph represents the occurrence of cells in each cluster. **f** UMAP analysis of cell senescence marker expression. The colors highlight different clusters. **g** Cells in cluster 9 are predominantly located in the Islets of Langerhans. H&E images are from an adjacent section of the iCLAP-IF-stained tissue.

## iCLAP-CyCIF uncovers P16 and 53BP1 associated pancreatic islet composition and function changes

Next, we showed that the iCLAP workflow could be seamlessly integrated with conventional multiplex immunostaining techniques, including CyCIF, IMC, and CODEX, enabling 40+ -plex detection in archival FFPE tissues (Fig. 4) while achieving high-sensitivity detection of low-abundance proteins. We demonstrated the application of iCLAP with CyCIF to co-label both senescence markers and key pancreatic tissue compartment markers, including endocrine markers such as insulin, glucagon, and somatostatin 28 (SST-28), as well as exocrine markers such as trypsin, lipase and carboxypeptidase A (CPA). (Fig. 4a).

Co-detection of senescence markers and islet functional markers (insulin, glucagon, and somatostatin 28) confirmed that the islet compartment exhibited high P16 and 53BP1-stained intensity signal. Our analysis revealed complex associations between senescence marker intensity and islet cell subtypes, showing that all three subtypes within pancreatic islets exhibited heterogeneous P16 expression, including both P16-positive and P16-negative populations (Supplementary Fig. 11a, b).

Next, we measured relative marker expression levels within individual islets (Fig. 4b). Correlation analysis of marker levels in individual islets revealed that high P16-expressing islets were associated with lower insulin and somatostatin 28 levels and higher glucagon levels (Fig. 4c, Supplementary Fig. 12a, b). Additionally, we observed a strong correlation between P16 intensity, insulin/glucagon intensity, and islet size, with larger islets exhibiting higher P16 intensity signal, higher glucagon intensity signal, and lower insulin intensity signal (Fig. 4d, e, Supplementary Fig. 12c–e). To further investigate the relationship between P16, 53BP1, and endocrine secretory factors at the single-cell level, we classified islet cells into four subtypes: Insulin+, Glucagon+, SST-28+, or negative for all three markers (Supplementary Fig. 12f, g). We also categorized individual cells based on their P16 or 53BP1 relative expression level (Supplementary Fig. 12h). Analysis of P16+ occurrence in islet cell subtypes showed that SST28+ cells were more likely to be P16+ compared to insulin+ and glucagon+ cells. Additionally, all three islet cell subtypes showed a high prevalence (> 90%) of 53BP1 expression (Fig. 4f,g).

We found that as the islet sectional area increased, there was a corresponding rise in the incidence of P16+ and Glucagon+ cells, while

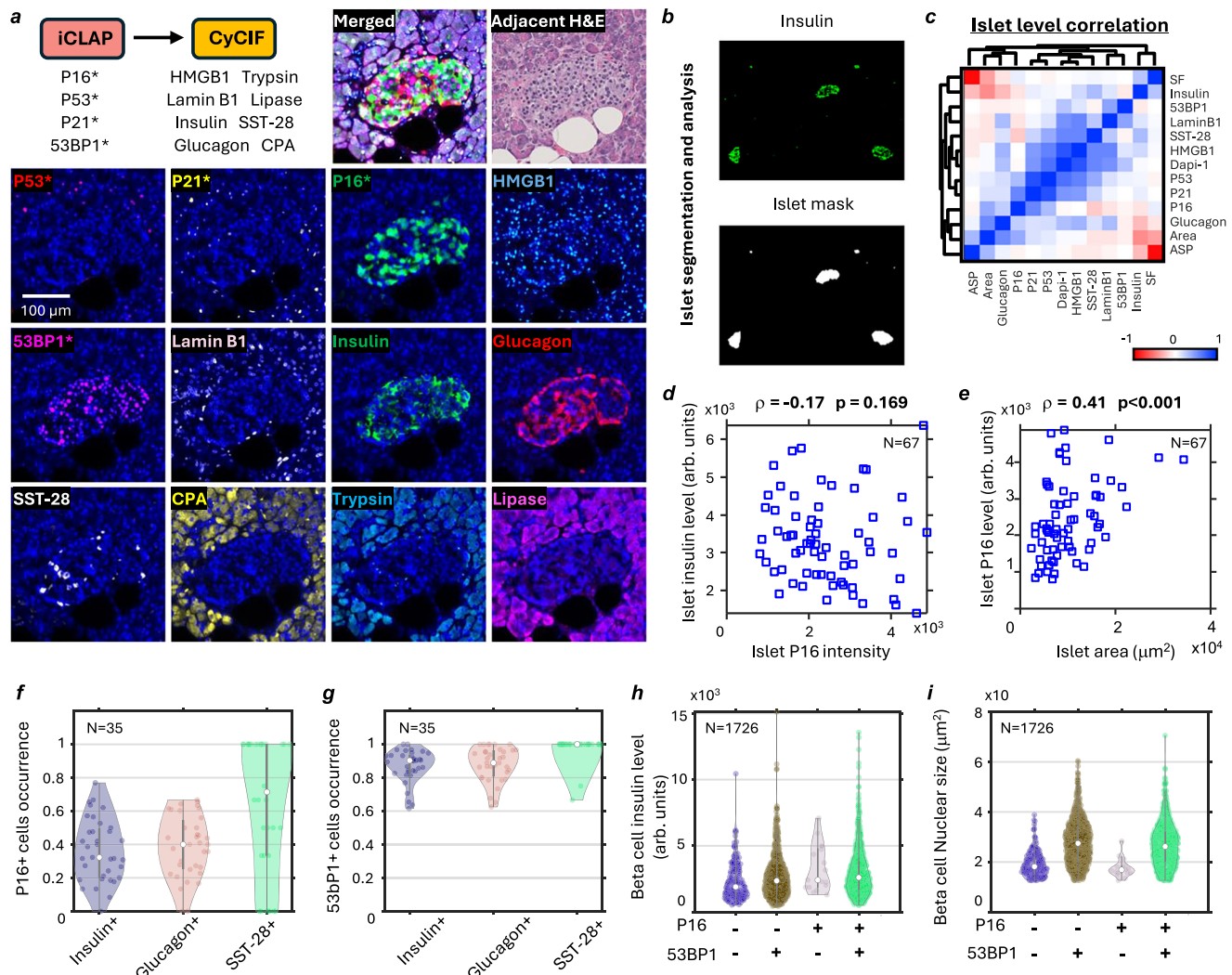

**Fig. 4 | Analysis of senescence and functional markers in pancreatic islets using multiplexed iCLAP-CyCIF. a** Using the iCLAP-CyCIF workflow to detect 12 markers in archival pancreatic tissue. The images show the detection of senescence markers (P16, P53, P21, 53BP1, Lamin B1, and HMGB1), as well as pancreatic cell markers (Insulin, Glucagon, Somatostatin 28, Lipase, Trypsin, and Carboxypeptidase A). **b** The stained islet cell markers (insulin and glucagon) were used to segment the islet region with image processing. **c** A heatmap illustrates the pairwise Pearson correlation coefficients rho (ρ) between senescence, pancreatic functional markers, and islet morphology at the islet level. Blue indicates positive correlations and red indicates negative correlations, as shown by the color scale (ranging from −1 to +1). Each variable is perfectly correlated with itself ($r = 1$) along the diagonal. **d, e** Scatter plots show islet-level correlations between P16 and insulin expression (in arbitrary units) (**d**) and between P16 expression (in arbitrary units) and islet area (**e**). Statistical analysis was performed using Pearson correlation (two-tailed). No significant correlation was observed between P16 and insulin expression ($r = -0.17$, $P = 0.169$),

whereas P16 expression was significantly correlated with islet area ($r = 0.41$, $P = 0.00057$). $N = 67$ islets. **f, g** Violin plots show the occurrence of P16-positive and 53BP1+ cells across different pancreatic cell types (Insulin+, Glucagon+, and SST-28+) in individual islets. The white dot indicates the median, and the grey vertical line indicates the interquartile range (25th–75th percentile). Sample sizes are indicated in each panel. Of the 67 islets identified, only islets containing all three endocrine subtypes (insulin+, glucagon+, and SST-28+ cells) were included for analysis ($n = 35$). **h, i** Violin plots show the beta cell insulin levels (in arbitrary units) (**h**) and nuclear size distribution (**i**) across different P16 and 53BP1 expression groups. The white dot indicates the median, and the grey vertical line indicates the interquartile range (25th–75th percentile). Sample sizes are indicated in each panel. Summary statistics, including mean, median, standard deviation (SD), standard error of the mean (SEM), minimum and maximum values, and 5th–95th percentiles, are provided in the Source Data files.

the incidence of 53BP1+ and Insulin+ cells decreased (Supplementary Fig. 13a–f). These findings indicate a strong association between islet size, cell subtypes, and senescence states. At the single-cell level, beta cells (Insulin+) expressing P16 or 53BP1 exhibited higher insulin signal (Fig. 4h, Supplementary Fig. 14a). The average insulin intensity levels for P16-/53BP1-, P16+/53BP1+, P16-/53BP1+, and P16+/53BP1+ cells were 2.34, 2.72, 3.30, and 3.05, respectively. Similarly, glucagon intensity and P16+/53BP1+ status in alpha cells (glucagon+) were correlated. Notably, glucagon intensity appeared more strongly correlated with 53BP1, showing a ~50% increase in expression in P16-/53BP1+ alpha cells compared to P16-/53BP1- cells. In contrast, glucagon intensity signal increased by ~30% in P16+ alpha cells with negative 53BP1

expression (Supplementary Fig. 14b, c). In contrast to the negative association between P16 and insulin observed at the islet level, single-cell analysis revealed a positive association between P16 and insulin relative expression. These findings suggest that the lower overall insulin expression in larger islets is primarily due to changes in islet cell subtype composition (i.e., loss of Insulin+ cells) rather than an increase in relative P16+ cells.

Cellular senescence is known to be associated with changes in morphology[42–47]. Here, we found that nucleus size increased with 53BP1 intensity level. The nuclear size of β cells with high 53BP1 intensity was ~50% larger than that of β cells with low 53BP1 intensity. In contrast, P16+ intensity was not correlated with nuclear size (Fig. 4i).

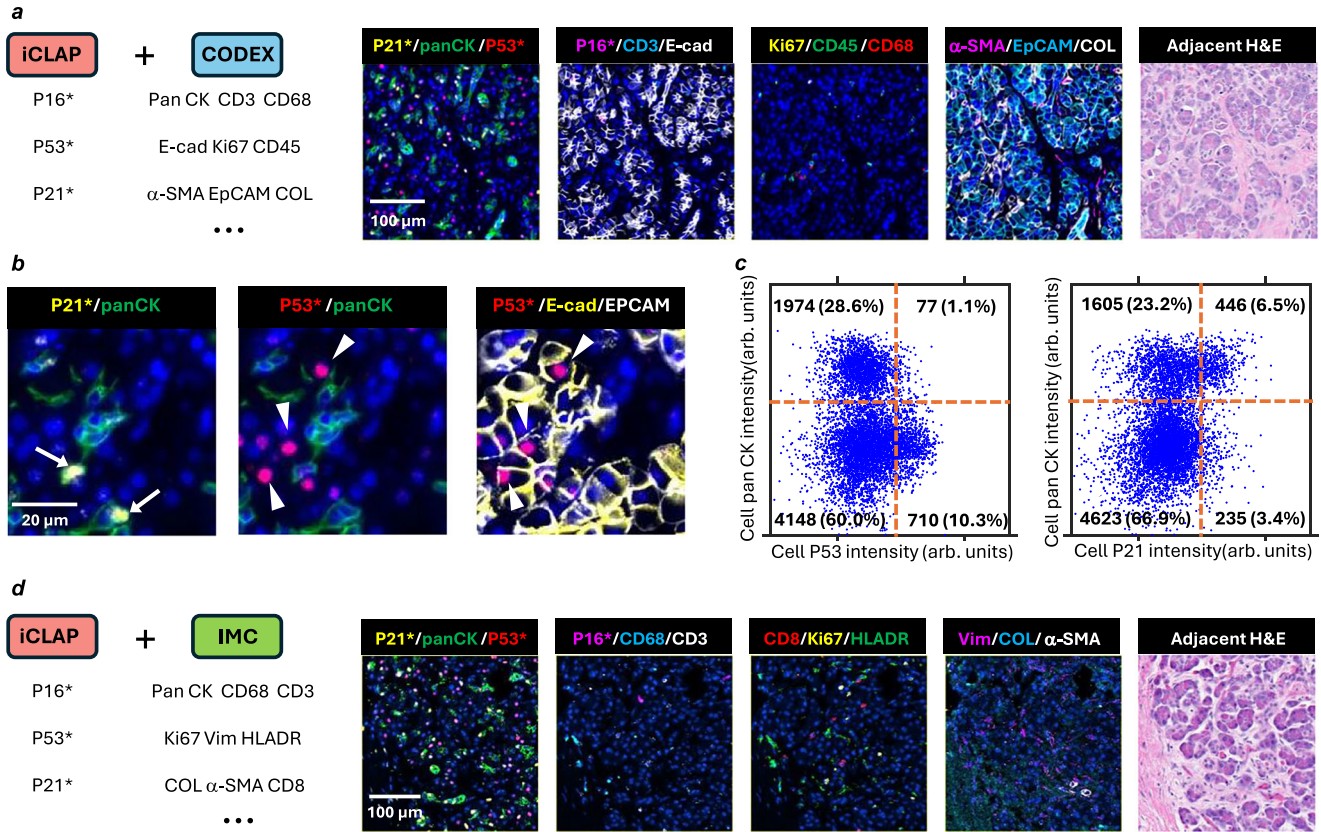

**Fig. 5 | Analysis of senescence and structural markers in pancreatic acini using multiplexed iCLAP-CODEX and iCLAP-IMC. a** Tissue imaging workflow using a 43-plex iCLAP-CODEX panel, composed of 3-plex iCLAP (P21, P53, P16) combined with a 40-plex CODEX antibody panel. Markers indicated with an asterisk (*) originate from iCLAP staining. Images display staining of P21*, P53*, P16*, panCK, E-cadherin, Ki67, CD45, α-SMA, and EpCAM in tissue sections, alongside an adjacent H&E-stained image. **b** Representative images illustrating that P21+ cells are typically panCK+, whereas P53+ cells are generally panCK-. **c** Quantitative analysis showing that 10% of P53+ cells are panCK+, while 65% of P21+ cells are panCK+, indicating distinct epithelial associations of these senescence markers; fluorescence intensities are reported in arbitrary units. **d** Tissue imaging using a 31-plex iCLAP-IMC panel, comprising a 3-plex iCLAP panel (P21, P53, P16) combined with a 28-plex IMC panel. Representative images highlight colocalization patterns across different markers. The staining results also highlight the spatial association of P21 and panCK.

Overall, our results suggest that the expression of senescence markers is closely linked to the insulin and glucagon expression.

### iCLAP-CODEX and iCLAP-IMC reveal distinct epithelial marker expression patterns of P21+ and P53+ cells in the acinar region

Next, we evaluated the marker profiles associated with P21+ and P53+ cells in the acinar compartment. The iCLAP-CyCIF analysis showed no significant differences in the intensity patterns of acinar enzymes (CPA, trypsin, and lipase) within regions exhibiting a high density of P21+ or P53+ cells (Supplementary Fig. 15a–h). Using iCLAP-CODEX, we simultaneously detected three senescence markers (P21, P53, and P16) alongside a 28-plex CODEX panel of tissue and cell markers, including panCK, E-cadherin, EpCAM, Ki67, α-SMA, collagen, CD3, CD45, and CD68 (Fig. 5a).

Our analysis revealed that P21+ and P53+ cells in the acinar compartment were distinct from immune cells and exhibited epithelial characteristics, as indicated by the absence of CD3, CD45, and CD68 staining and the presence of E-cadherin/EpCAM staining (Supplementary Fig. 16a–d). Notably, P21+ cells more frequently co-expressed panCK, whereas P53+ cells were typically panCK negative (Fig. 5b). This co-expression pattern was consistently reproduced in two additional patient samples from old donors (50 yr+), in which panCK+ cells exhibited a significantly higher occurrence of P21+ labeling compared to P53+ (Fig. 5c, Supplementary Fig. 17a–c). Quantitative analysis further showed that panCK+ cells displayed a significantly higher occurrence of P21+ labeling than P53+ across all three samples (Fig. 5c, Supplementary Fig. 17a–c).

Finally, we demonstrated the integrability of iCLAP with IMC, showing that senescence markers can be detected alongside 40 tissue markers mapped using IMC (Fig. 5d). The IMC dataset corroborated the higher frequency of panCK co-expression among P21+ cells. Staining by iCLAP-IMC and iCLAP-CODEX was validated by co-expression of canonical markers within each platform (Supplementary Fig. 18a–e). Compared with conventional IMC, iCLAP-IMC showed comparable intensity distributions and spatial patterns across the 40-plex panel (Supplementary Fig. 19a–f). Together, these results demonstrate the usefulness of integrating iCLAP with high-plex multiplexing methods, underscoring both the robustness and the broad applicability of the iCLAP workflow across CyCIF, CODEX, and IMC platforms.

## Discussion

The detection of low-abundance proteins in formalin-fixed, paraffin-embedded (FFPE) tissues has long been challenging due to antigen retrieval issues and signal loss, leading researchers to rely on frozen tissues for improved antigenicity[48–50]. However, FFPE-archived tissues are significantly more accessible than frozen samples and achieving high-sensitivity antigen/protein detection is of central importance[51–54]. Here, we demonstrate that the proposed iCLAP method enables both high-sensitivity and high-plex detection while preserving FFPE tissue architecture. This advancement expands the applicability of FFPE tissues for biomarker studies and translational research, providing access to vast clinical tissue repositories without compromising detection sensitivity.

The iCLAP method utilizes TSA-based fluorescence for high-sensitivity antigen detection. However, the resulting fluorescence signals are often significantly stronger than those produced by fluorophore-conjugated antibodies. Therefore, effective fluorophore removal is crucial to re-stain the same tissue section. Our results demonstrate that iCLAP achieves superior fluorophore removal efficiency in TSA-based immunofluorescence staining compared to CyCIF methods[5], enabling the re-use of the same sections for multiplexed analysis. In iCLAP, low-abundance proteins are detected through TSA staining, where only a single antigen is stained in each cycle, followed by antibody removal. As a result, each antigen staining cycle is time-consuming (typically ~24 h), which may present a practical limitation in achieving a high level of multiplexing when detecting low-abundance proteins within the same tissue.

Our study demonstrates that iCLAP effectively co-detects key senescence protein markers alongside tissue structural and functional markers, as well as cell-type-specific markers for in-depth analysis of cellular senescence in situ. Our analysis of senescence marker intensity reveals that these markers are highly associated with specific tissue compartments in the pancreas. Importantly, we found that the co-expression of multiple senescence markers is not a common phenotype in the analyzed human pancreatic specimens from older adults. This finding contrasts sharply with observations from in vitro cell models of irradiation-induced senescence and replicative senescence[55–59], where the expression of multiple senescence markers is typically increased. These results suggest that the process of cellular senescence in human tissues in situ may be regulated differently from in vitro models, likely due to the distinct microenvironment within tissues.

Beyond senescence analysis, iCLAP's multiplexing capability and sensitivity for low-abundance proteins make it a powerful tool for studying various critical biological processes in situ. We demonstrated that iCLAP enables sensitive detection of transcription factors and secreted proteins in archival tissue sections. By integrating iCLAP with other high-plex immunolabeling methods, it facilitates the spatial analysis of transcriptional activity, cytokine/chemokine gradients, and their interactions within fully mapped tissue and cellular components. Thus, we anticipate that the iCLAP workflow can be expanded to study a wide range of biological processes in situ, including stem cell maintenance and differentiation[60–62], cancer immune evasion[63–65], and senescence-associated secretory phenotypes[66–68] processes often linked to differentially regulated transcription factors or secreted factors.

## Methods

### Ethics statement
This study was conducted in accordance with all relevant ethical regulations. Human tissue specimens were obtained from archival pathology collections at Johns Hopkins University and from the Joslin Diabetes Centre, Harvard Medical School, with approval from the respective institutional review boards. All tissue samples were collected with informed consent obtained from patients at the time of surgery for research use of tissue, and were de-identified prior to analysis.

### Tissues and specimens
The tissue blocks were sectioned at 4 µm onto plus slides by the Johns Hopkins Oncology Tissue Service core for immunostaining and H&E staining. Unstained tissue microarray sections from healthy tissues and tumours across various organs, including the breast (BR1008b), liver (LV242a), cervix (CR501a), skin (ME481e), ovary (OV488b), and pancreas, were obtained from TissueArray.com (Derwood). All stained H&E sections were scanned using a Hamamatsu S210 Digital slide scanner (Hamamatsu) with a 20× objective. Details of all human specimens, including sample identifiers, donor age and sex, section

numbers, and corresponding figure panels, are provided in Supplementary Data 1.

### Antibodies
Detailed information on the antibodies used in this study, including their sources, clones/catalog numbers, and dilutions, is provided in Supplementary Data 2.

### Tissue processing
Four-micron paraffin sections were baked at 42 °C for 3 h and dried overnight at room temperature with a desiccator, dewaxed using xylene, rehydrated with a series of alcohols, and concluding with several times of dipping in water. The tissue slides were transferred to a heat-resistant plastic bowl filled with antigen retrieval solution (Vector laboratories, H-3300-250) and subjected to 20 min of heating in a food steamer (Bella).

### Immunofluorescence staining
A primary antibody was applied, and tissue slides were incubated for an optimized time at room temperature or at 4 °C overnight. Slides were then incubated with a secondary antibody conjugated with a fluorophore based on the host species of the primary antibody. Donkey anti-Rabbit IgG (H + L) Highly Cross-Adsorbed Secondary Antibody, Alexa Fluor™ 647 (ThermoFisher, A-31573) were used for rabbit host primary antibody, and Goat anti-Mouse IgG (H + L) Highly Cross-Adsorbed Secondary Antibody, Alexa Fluor™ 647 were used for mouse host primary antibody (ThermoFisher, A-21236).

### iCLAP workflow
Low-abundance proteins or antigens were first labeled through sequential TSA-based primary antibodies. Three antigens were typically labeled within each imaging cycle. Tissue sections were blocked with Blocker™ casein (Thermo Scientific, 37528) for 15 min, followed by 2-min incubation with TrueBlack Lipofuscin Autofluorescence Quencher (Biotium, 23012). For each antigen detection, the primary antibody was diluted in the Blocker™ casein blocking solution, and tissue slides were incubated for 40 min at room temperature or at 4 °C overnight. Slides were then incubated for 8 min with 100 µl of anti-mouse HQ (Roche Diagnostic, 760-4814) or anti-rabbit HQ (Roche Diagnostic, 760-4815), depending on the host animal of the primary antibody. Tissue slides were then incubated with 100 µl of anti HQ-HRP (Roche diagnostic, 760-4820) for typically 6 min. Tissue slides were then incubated with one of TSA plus Opal dyes (Fluorescein, Cy3 or Cy5 from Akoya; NEL741001KT, NEL744001KT, NEL745001KT) for 10 min. The VectaPlex™ Antibody Removal Kit (Vector Laboratories, VRK-1000) was then applied to strip antibodies in sections. To be noted, heat-induced epitope retrieval (HIER) was performed only once at the start of the staining series, whereas subsequent antibody removal was achieved using a chemical elution kit (Vector Laboratories, VRK-1000). Blocking, antibody incubation and labeling steps were then repeated for the remaining antigens to be detected, until all three antigens in this imaging cycle were labeled with Fluorescein, Cy3, and Cy5.

Counterstaining was performed with 0.6 µM Hoechst 33342 in Blocker™ casein solution for 15 min. The stained tissue sections were then imaged using an inverted fluorescence microscope (see detailed in the Fluorescence Microscopy section). One drop of TBS-T was added to the tissue region to avoid evaporation while imaging. After imaging, the fluorophore inactivation steps were performed to reduce the fluorescence signal to a background level. Tissue sections were placed in a transparent box, which was then filled with the bleaching solution containing 2 M $H_2O_2$ and 3 mM EDTA in PBS at pH 12.5. The transparent box, holding the tissue slides and bleaching solution, is positioned between two 5000 lux light pads (HSK, 615517997868) for one hour to facilitate fluorophore inactivation. Slides were then rinsed

and blocked prior to the next staining round. All TSA-based detections were performed prior to incorporation with other detection methods.

## Fluorescence microscopy

Fluorescently labelled tissue sections were imaged with a Hamamatsu Flash 4.0 CMOS camera mounted on an inverted research microscope (Ti-E, Nikon). The microscope is equipped with a motorized stage and motorized excitation and emission filters controlled by NIS-Elements (Nikon). Lumencor SpectraX 6 (Lumencor) was used as the light source. For each sample, a custom grid setup was determined to acquire images covering the entire tissue area using an S Fluor 10x microscope objective with an NA of 0.5 (MRF00100, Nikon). For image stitching, the grid step size is set to contain a 10% overlap between adjacent images. The Perfect Focus System (Nikon) was used to maintain a consistent imaging focal plane across the tissue area. Under this microscopic setup, the pixel size of the acquired images was 0.65 μm. The images acquired in each grid were stitched using a previously described method[69,70].

## Tissue image registration

To spatially align immunolabeled images derived from different rounds of microscopy images or between different imaging modalities (i.e. IF and CODEX, IF and IMC), we used a previously established registration pipeline[71,72]. Our registration method is based on images of nuclei. The DAPI channel or equivalent is used to represent the nuclei images. Our registration consists of two steps: global rigid registration and local grid-based deformative registration[72]. To align images acquired from different modalities, they were first rescaled to the same pixel size. The global rigid registration was applied to the downsampled images to improve computational efficiency. The local deformative registration was implemented using full-resolution images to ensure high registration accuracy. The grid step size for the implementation of local deformative registration was 500 pixels. Aligned whole slide images were then output into ome-tiff format using the libvips library[73], and qualitatively examined using Qupath[74].

## Cell profiling and analysis in immunolabeled images

Nuclei segmentation in the tissue images was first performed on the DAPI-stained channel using the StarDist pre-trained model[75]. The cell boundary for each segmented nucleus was defined as an expansion of 4.5 μm (7 pixels) from the nucleus boundary. If this expanded boundary overlapped with a neighbouring nucleus, the boundary was set at the midpoint between the two nuclei. Image processing for quantification of cellular morphological features from fluorescent images was carried out using a custom program developed in MATLAB[69,76] (Mathworks, MA). Morphology features, including area, aspect ratio, circularity, equivalent radius for detected nuclei and cells, were measured using a previously established method[69,70,76]. Nuclei Intensity features, including mean and total intensity, were measured across all detected and aligned channels after background removal. Background images for each channel were generated using a 2D median filter with a $7 \times 7$ pixel window on images down-sized by a factor of 10, then rescaled to the original image size.

To quantitatively classify senescence expression profiles, we first selected 11 representative $2000 \times 2000$ pixels ROIs in the whole slide images, avoiding regions that exhibited staining artifacts such as tissue folding. The mean intensity from cell nuclei from stained markers was used to represent senescence expression profiles, except for P16, where cell area was used due to its cytosolic staining pattern. K-means clustering analysis was then applied to identify 20 expression subtypes based on detected senescence markers. Clustering was performed using reduced intensity features that captured 95% of the variance from Principal Component Analysis (PCA), with z-scored normalized intensity features. Uniform manifold approximation and projection

(UMAP)[77] was used to visualize the expression landscape based on PCA-reduced features. The analysis was performed using a custom program developed in MATLAB.

## Pancreatic islet segmentation and islet cell subtyping

To segment the pancreatic islets, intensity thresholds were manually set to detect positively stained regions in insulin- and glucagon-labeled images. A binary islet map was generated by identifying areas positive for either insulin or glucagon. The resulting islet maps were refined using morphological closing and opening operations, followed by filling non-positive pixels that are enclosed within the positively stained islet regions. Stained islet regions smaller than 400 μm² were excluded from further analysis. A previously established custom program[69] was used to extract intensity and morphological features of the identified individual islets. The extracted features included islet area, aspect ratio, shape factors, average intensity, and total intensity of the stained channels.

## iCLAP integrated workflow with CyCIF multiplexing

CyCIF staining was performed using an updated version[5]. In short, the tissue section was incubated with fluorophore-conjugated primary antibodies at 4 °C for 12 h. Typically, three different antibodies with distinct fluorophores were applied simultaneously. DAPI was used to counterstain the nuclei. The stained tissue sections were then imaged using a fluorescence microscope to capture the entire tissue region across all labeled fluorophore channels. The fluorescence signals were subsequently deactivated using the iCLAP fluorophore inactivation protocol. The slides were then ready for the next round of staining and imaging with the designated fluorophore-conjugated primary antibodies.

## iCLAP workflow integrated with Imaging mass cytometry (IMC)

Sections were blocked with 3% BSA at room temperature for 45 min, then incubated overnight at 4 °C in the IMC antibody cocktail solution. For nuclear counterstaining, sections were exposed to Cell-ID Intercalator-Ir (Standard Biotools) diluted 1:400 in Maxpar PBS for 30 min at room temperature. Tissues were further counterstained with 0.5% ruthenium tetroxide (Electron Microscopy Sciences PN 20700-05) at a 1:2000 dilution for 3 min at room temperature. Following Maxpar water washes, slides were air-dried and placed into the Hyperion Imaging Plus System (Standard Biotools) for image acquisition at the Johns Hopkins Mass Cytometry Facility.

## iCLAP workflow integrated with CODEX multiplexing

Tissue slides were incubated in pH 9 antigen retrieval buffer at 95 °C for 40 min. Following this, the slides were processed according to the PhenoCycler-Fusion manual, including a photo-bleaching step. The photo-bleaching was performed in buffer (25 mL PBS, 4.5 mL 30% $H_2O_2$, 0.8 mL NaOH) twice, each for 45 min.

## Statistical analysis

Statistical analysis was performed using two-way analysis of variance (ANOVA) to assess differences in fluorescence intensity across the iCLAP staining/bleaching cycle and imaging channel on fluorescence intensity (Fig. 1d). Tukey's multiple comparisons test was applied post hoc to evaluate group differences. GraphPad Prism (version 9.5.1) was used to determine statistical significance. Data are presented as mean ± standard deviation (SD) from 5 biological replicates. Results were considered significant at $P < 0.05$ (*), $P < 0.01$ (**), and $P < 0.001$ (***), $P < 0.0001$ (****). "ns" indicates no significant difference.

## Reporting summary

Further information on research design is available in the Nature Portfolio Reporting Summary linked to this article.

## Data availability

The processed imaging-derived quantitative data generated in this study, together with representative regions of interest (ROIs) extracted from multiplexed whole-slide images, have been deposited in **Zenodo** (https://doi.org/10.5281/zenodo.18202750, https://doi.org/10.5281/zenodo.18202615, https://doi.org/10.5281/zenodo.18202404, and https://doi.org/10.5281/zenodo.18202295). These data constitute the minimum dataset required to interpret, verify, and extend the findings of this study. Supplementary Data 1 provides human specimen metadata and figure-to-sample mapping necessary for interpretation of the imaging results. Supplementary Data 2 lists all antibodies used for iCLAP immunolabeling, including targets, clones, suppliers, and dilutions. Source Data supporting the quantitative analyses are provided with this article. Raw multiplexed whole-slide imaging data generated from human FFPE tissue sections are not publicly available due to the large size of the datasets and ethical and privacy restrictions associated with human tissue donors and institutional data governance policies. Access to raw whole-slide images may be considered under restricted access for non-commercial research purposes. Requests for access should be directed to the corresponding author and will be reviewed on a case-by-case basis in accordance with institutional policies. Source data are provided with this paper.

## Code availability

Custom scripts used for image analysis in this study were developed by the authors. Code necessary to reproduce the analyses presented in this manuscript, including ROI-based quantification and statistical analysis, has been deposited alongside the associated datasets in Zenodo (https://doi.org/10.5281/zenodo.18203368).

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

## Acknowledgements

The authors thank Xuan Yuan for her contributions to imaging mass cytometry data acquisition at the Mass Cytometry Facility at Johns Hopkins University. This work was supported by the NIH Senescence Network (SenNet) through grants UG3CA275681 and UH3CA275681(PHW), which provided substantial funding for the development of this study. The authors acknowledge the following additional sources of support: U54AR081774 (DW); U54CA268083 (DW); R01CA300052 (DW), U54AG075932 (BS), P30CA006973 (WJH), S10OD034407 (WJH), and 1U54AG075941 (GAK) all from the National Institutes of Health.

## Author contributions

F.W., P.H.W., and D.W. conceived and designed the study, F.W. and S.Y.Z. developed the iCLAP bleaching method, and conducted the experiments. F.W., Y.N.C., P.J.Y., M.J.K., S.L., G.K., S.P., and G.P. contributed to experimental optimization and data acquisition. R.H.Y., B.F.Y., and K.S.H. assisted with image analysis. Q.F.Z. and R.A.A. contributed to the CODEX data acquisition and interpretation. S.M.S., C.C., and W.J.H. contributed to the IMC data acquisition and interpretation. P.R., K.I., C.A.M., N.M., and G.A.K. contributed to the collection and transport of the additional tissue cohort. B.S., L.W., W.J.H., R.A.A., D.W., and P.H.W. provided critical feedback on the methodology. F.W., P.H.W., and D.W. contributed to manuscript writing and editing. DW and PHW supervised the study, secured funding, and provided overall guidance. All authors reviewed and approved the final manuscript.

## Competing interests

The authors declare no competing interests.
