## [Transparent Peer Review file · Nature Communications]

iCLAP: a novel method for integrable co-detection of low-abundance antigens with high-plex immunostaining

Corresponding Author: Dr Pei Hsun Wu

Version 0:

Reviewer comments:

Reviewer #1

(Remarks to the Author)

This article describes a novel method for the multiplexed protein detection of low abundance antigens in FFPE tissue, which is mainly attributed to an optimized bleaching step. I have a few minor comments which I have detailed below:

- 1) "The TSA-based immunofluorescence staining of P53 and P16 were further validated with IHC staining using antibodies from multiple sources targeting the same antigen (Supp. Fig. 2 a-b)." This supplementary figure also includes P21, please revise the text accordingly.
- 2) "For P53 and P21, clear true positive staining signals were not observed for traditional IF staining, even with extended antibody incubation times and increased antibody concentrations (Fig. 2d-f and Supp. Fig. 3a-b)." Text should read Fig. 2d-e instead of 2d-f.
- 3) "Notably, HMGB1 and Lamin B1 exhibited adequate staining with traditional immunofluorescence." Were these the only 2 that worked with traditional IF? How many out of the 29 antibodies work with IF?
- 4) Figure 3a-c: there is discrepancy in the text and the figure between which markers were stained in the 1st and 2nd rounds. Was 53BP1 stained in the 1st or 2nd round? I don't see any LMNB1 staining in 3c.
- 5) "Stained images readily showed that cells in acinar regions with high expression of P21 and P53 originated from different cells (Fig. 3c)." This is quite hard to see, even when zooming in. Perhaps consider including a zoomed in region that supports this statement.
- 6) "Next, we showed that the iCLAP workflow could be seamlessly integrated with conventional multiplex immunostaining techniques, including CyCIF, IMC, and CODEX, enabling 40+ plex detections in archival FFPE tissues (Fig. 4), while achieving high sensitivity detection on low abundance proteins." Only 12 markers are shown. IF 40+ markers have not been tested, revise statement.
- 7) There is no mention of figure 4d or Supp. Fig. 10e-f in the text.

Reviewer #2

(Remarks to the Author)

The focus of the manuscript of Wu and colleagues is to include the detection of low abundance proteins in established protocols using multiplex techniques for protein detection. To achieve this the three multiplex approaches CyCIF, CODEX, and IMC are used and combined with a tyramide signal amplification and bleaching workflow termed iCLAP. The novelty, the authors report, is to improve the bleaching procedure to get several rounds of tyramide signal amplification. In combination with the multiplex approaches it is claimed that a cellular landscape of proteins can be visualized in combination with barely expressed proteins.

The approach is technically interesting as it extends the number of proteins to be analyzed in tissue samples. However, many issues remain unclear. Among various methodological points the most important one is that the authors want to investigate cellular senescence, especially in pancreatic tissue. A lot of data is collected, but in the end, it is not clear which in-depth analysis of cellular senescence the authors were able to reveal by their approach. Therefore, the manuscript needs to be substantially revised and improved.

Major comments:

Major comment 1:

The reader should get a better overview about the material used in this study. As senescence markers are in the focus of this

work, and as age might influence the quality and structure of the tissue slices, it is important to get more information about the data shown. In the methods the authors mention that grossly normal pancreatic tissue of pancreatectomy specimens was used. Figure 3 shows data of a sample derived from a 63-year old donor. In figure 2, 4, and 5, where senescence markers and many other parameters were investigated and compared, and in the supplementary figures, we do not get to know anything about the samples. This has to be changed throughout and discussed adequately.

Major comment 2:

The TSA-technique was used for p16, p21, p53, and 53BP1 in the data presented in Figure 3. The signal-to-background-ratio looks quite well, however, a signal is only visible for p16 and 53BP1. I would have expected positive signals for p21 and Lamin B1 somewhere in the slice as well. Is the same patient donor for the data presented in Suppl. Fig. 5 (where p21/Lamin B-positive cells are visible)? In the left part of Fig. 3c there are pink dots in the acinar tissue. Is this corresponding to p53 (indexed in red)? If yes, please, change the colours. Why did the authors decide to use the TSA technique for p16? This is also easy to detect by classical IF and I would say p16 is not a low abundant protein in islet cells.

Major comment 3:

Starting at Figure 3d the authors focus on a detailed analysis of the whole pancreatic parenchyma. It remains unclear, where the 70,000 individual cells are from (proportion of tissue covered by exocrine vs. that covered by endocrine cells) and which selection parameters have been used. Is it one area (the one marked in part g of Fig. 3) or are different areas with more than one islet analyzed? How many islet cells were included in the 70,000 individual cells?

Major comment 4:

Certainly, reproducibility is of great interest for the message of the paper. Are these clusters described in Fig. 4 reliably identified in samples of different donors? Which of the clusters are age-dependent? Can anything be said about gender differences? As Dapi is included here, we can see a huge variation in the nuclear staining in the cells. I would have expected that the Dapi signal is stable throughout and I would interpret a lower Dapi signal as a generally lower quality of the staining (maybe because of the quality of the tissue slice or else). How do you explain this variation?

For the analysis of acinar cells with iCLAP-CODEX it is also of great interest, if the correlation p21+/ductal component and p53+/acinar cells can be reproduced in different samples and is (in)dependent of the age of the donor, gender etc..

Comparisons between samples of different donors addressing the issues mentioned above should be added.

Major comment 5:

It is under general discussion if one can really conclude from the intensity of a fluorescent antibody to the level protein expression. Fluorescence signals are quite difficult to standardize to protein expression. I would recommend to be more careful regarding this issue and to change the wording throughout the manuscript to "fluorescence intensity levels" or something equivalent.

Major comment 6:

In the last part of the results with iCLAP-CyCIF the authors say that "Overall, our results suggest that the expression of senescence markers is closely linked to the secretion of islet functional factors.". The manuscript contains no functional data. It is "just" fluorescence signals indicating the existence of protein expression. Positivity for p16 and 53BP1 is associated with higher intensities for insulin and glucagon fluorescence staining. This might indicate more insulin/glucagon within the cell (although I would be very cautious about this, as we have archived embedded tissue), but it says nothing about secretion.

Minor comments

- (a) Suppl. Fig. 1: I assume two consecutive slices of the same sample were used for comparing the different methods, right? Please mention this in the figure legend. Please also comment on why the bleaching procedure has no influence at all on the blue nuclear Dapi staining.
- (b) Suppl. Fig. 1b, right part: Please reformat the bars and sizes in the far right row. There are some overlaps.
- (c) Suppl. Fig. 3: Please define the type of cells that was analysed in these images.
- (d) Suppl. Table 1: In the text it is said that 29 antibodies were tested and Suppl. Table 1 is given as a reference. I assume the sentence refers to those antibodies where it is indicated that TSA was used, but this are only 26. Please comment.
- (e) What does the index "1" for Dapi mean in Fig. 4c?
- (f) Statistical analysis: I assume that the analysis describes the data presented in Fig. 1d. Please add this information, and also add if any other statistical evaluation was performed.

Reviewer #3

(Remarks to the Author)

The authors present integrable Co-detection of Low Abundant Proteins (iCLAP) which uses tyramide signal amplification (TSA) followed by chemical bleaching to integrate detection of lowly-expressed antigens via enzymatic deposition of tyramide molecules with highly-multiplex imaging techniques CyCIF, CODEX and IMC. The authors demonstrate complete fluorophore bleaching using their novel method, improved sensitivity over conventional IF, and integration with other multiplex methods. For example, integration with CyCIF enabled islet cell typing with insulin, glucagon and somatostatin 28 and revealed a negative correlation with islet size/P16 expression and insulin expression.

The innovation in this work is the novel fluorescence bleaching method which is 2M H₂O₂ (almost 7%) and 3mM EDTA in

PBS at pH 12.5. Lin et al. previously published a bleaching method using 4.5% H₂O₂ and 24 mM NaOH made up in PBS, and others have demonstrated 20 mM NaOH and 3% H₂O₂ in PBS, pH 7.4 [PMID: 35545666]. Both works demonstrated 70-80% tissue loss over 10 rounds using these conditions. The authors should quantify tissue loss in the iCLAP protocol, especially in normal tissues which tend to be more fragile in the two above-mentioned references. Tissue loss could be especially likely with the additional rounds of antigen retrieval inherent in the TSA protocol.

Pulsawatdi et al. demonstrated that antigenicity changes after multiple rounds of HIER, and Eng et al. showed that repeated rounds of H₂O₂ quenching and blocking also alter non-specific background and dynamic range. Quantification of these metrics for iCLAP versus single plex would be informative. For example, in supplemental figure 5, LaminB1 staining looks dimmer in iCLAP-IF than in single plex but there is no quantification. Finally, reproducibility is an open question. While it had been addressed in each technology individually [PMID: 34266881, 35545666] I believe the authors should quantify the reproducibility of combining the TSA and (CyC)IF methods into iCLAP.

Overall, the authors report a useful method to combine well-established TSA protocols and multiplex imaging methods (IF, CyCIF, CODEX). As they state, iCLAP could be a “powerful tool for studying various critical biological processes in situ” however, additional quantitative metrics for tissue loss, SNR compared with single plex, and reproducibility would be informative for researchers wishing to adopt this method.

Specific comments:

1. Figure 1: Quantify any tissue loss due to bleaching and antigen retrieval. This should be done on the tissue microarrays containing normal breast and skin for best evaluation.
2. Figure 2: Quantify SNR of conventional IF p16.
3. Supplemental 5: quantify signal to noise ratio of single plex and iCLAP-IF.
4. Figure 3: Quantify the reproducibility of the iCLAP-IF method across adjacent slides. Also, clustering the heatmap in figure 3e would improve readability, especially with a dendrogram to show the similarity of senescence marker clusters as shown on the umap in f.
5. Supplemental 7: how much, if any tissue loss was experienced in these TMAs? Normal breast and normal skin have shown particular fragility in other platforms.
6. In figure 4, please report the p-value as well as rho for the Pearson correlation.
7. Supplemental figure 9: add p-values on Pearson correlations, add Pearson correlation for f-g.
8. Supplemental figure 10c is missing the x-axis label. 53BP1 representative images should be shown (quantified in b).
9. Supplemental figure 12 shows no quantification to support the lack of association between P21, P53 and acinar enzyme expression. Unsupervised clustering on single cell mean intensity would support this claim.
10. The authors state regarding Figure 5 c that “P53+ cells arise from acinar cells”. I don’t agree that lack of panCK in P53+ cells means they have a different cell of origin. This statement and the section title “distinct cellular origins of P21+ and P53+ cells in the acinar region” should be replaced with a less strong claim.
11. Figure 5 d lacks validation of IMC staining compared to standard IMC. For example, while Vim and collagen seem to match the H&E, the staining seems weak and diffuse. The iCLAP + IMC data should be quantitatively compared to standard IMC.
12. Supplemental figure 13 would benefit from quantification showing P53+ and P21+ cells are immune marker negative.

In summary, beautiful staining of low abundance targets. As a potential user, I wonder if the authors could comment on the difficulty of optimizing panel order. Are there any lessons learned during the development of this method?

Version 1:

Reviewer comments:

Reviewer #1

(Remarks to the Author)

Thank you to the authors for addressing my comments. I have no further remarks.

Reviewer #2

(Remarks to the Author)

In the revised manuscript of Wu and colleagues my issues are addressed adequately.

During re-evaluation of my former major comment 6, I recognized that the text box marked as revised text does not fully match with the text in the revised manuscript (lines 209 to 295). I assume something is wrong with the copied part in the answer-to-referees-letter. As the altered text in the manuscript is o.k., it doesn’t matter, I was just wondering, what happened.

Two minor points that should be changed:

1. I recommend to add scale bars to all panels shown in the Supplementary Data (missing e.g. in the new version of Suppl. Fig. 9 and in Suppl. Fig. 15).
2. Fig. 3c and new Suppl. Fig. 7a and b: Please change „um“ to “ μm ”.

Reviewer #3

(Remarks to the Author)

The authors have satisfactorily addressed all of my previous concerns, including tissue retention, signal to noise ratio, reproducibility and statistics and the manuscript is now suitable for publication.

We thank the reviewers for their thoughtful and constructive feedback, which has helped us to significantly improve the manuscript. Below, we provide detailed, point-by-point responses to all comments. Revisions made in the manuscript are highlighted in red text.

REVIEWER COMMENTS

Reviewer #1 (Remarks to the Author):

This article describes a novel method for the multiplexed protein detection of low abundance antigens in FFPE tissue, which is mainly attributed to an optimized bleaching step. I have a few minor comments which I have detailed below:

We thank Reviewer #1 for the positive evaluation and thoughtful feedback. We have addressed each point thoroughly and revised the manuscript based on the reviewer's suggestions accordingly.

1) "The TSA-based immunofluorescence staining of P53 and P16 were further validated with IHC staining using antibodies from multiple sources targeting the same antigen (Supp. Fig. 2 a-b)." This supplementary figure also includes P21, please revise the text accordingly.

We thank the reviewer for pointing this out. We have revised the text to accurately reflect the content of Supplementary Figure 3, which includes validation data for P21 in addition to P53 and P16. The manuscript is updated; accordingly, the sentence now reads:

*"The TSA-based immunofluorescence staining of P53, P16, and P21 were further validated with IHC staining using antibodies from multiple sources targeting the same antigen (**Supp. Fig. 3 a-b**)."*

2) "For P53 and P21, clear true positive staining signals were not observed for traditional IF staining, even with extended antibody incubation times and increased antibody concentrations (Fig. 2d-f and Supp. Fig. 3a-b)." Text should read Fig. 2d-e instead of 2d-f.

We really appreciate the reviewer's help in catching this oversight. We have corrected the figure reference in the main text accordingly.

3) "Notably, HMGB1 and Lamin B1 exhibited adequate staining with traditional immunofluorescence." Were these the only 2 that worked with traditional IF? How many out of the 29 antibodies work with IF?

We appreciate the reviewer's insightful question. In this work we tested a total of 29 senescence-related antibodies for 18 antigens. Among these antibodies tested, four antibodies demonstrated reliable staining using standard IF methods. These include both the primary and conjugated forms of Lamin B1 and HMGB1 antibodies. All other senescence markers in our panel required signal amplification to achieve detectable signal, underscoring the importance of the iCLAP approach for profiling low-abundance targets in FFPE tissues.

To clarify this point, we have updated the antibody list (Supp. Table 2) by adding two new columns: "Application" and "Performance with IF." The Application column indicates whether each antibody was considered as a senescence marker or a structural marker, while the Performance with IF column provides qualitative assessments of signal strength using conventional immunofluorescence, categorized as strong, weak, not detectable, or not evaluated.

4) Figure 3a-c: there is discrepancy in the text and the figure between which markers were stained in the 1st and 2nd rounds. Was 53BP1 stained in the 1st or 2nd round? I don't see any LMNB1 staining in 3c.

We apologize for the confusion. 53BP1 was stained in the second round using the TSA method. In the first round, P16, P53, and P21 were stained using TSA and imaged. Following this, the iCLAP bleaching protocol was applied, and 53BP1 was subsequently stained in the second round. Finally, Lamin B1 and HMGB1 were stained using conventional immunofluorescence.

To address this more clearly, we have updated the Fig. 3a caption to reflect both staining order and detection methods. The revised caption now reads:

“Figure 3. Multiplex senescence marker detection with CLAP-IF reveals subpopulation of cells with distinct marker expression patterns in a human pancreas. a. Schematic of the 6-plex iCLAP-IF workflow for senescence marker detection. P16, P53, and P21 were stained and imaged in the first round, while 53BP1, HMGB1, and Lamin B1 were detected in the second round. TSA was used for P16, P53, P21, and 53BP1 detection, whereas HMGB1 and Lamin B1 were visualized using conventional immunofluorescence.”

We thank the reviewer for pointing out the invisible LMNB and we realized the LMNB channel was not included in the images. We have regenerated the merged images to add the LMNB1 channel. The updated **Fig. 3c** is provided below:

5) "Stained images readily showed that cells in acinar regions with high expression of P21 and P53 originated from different cells (Fig. 3c)." This is quite hard to see, even when zooming in. Perhaps consider including a zoomed in region that supports this statement.

To address this, we have added Supp. Fig. 7 (reproduced below). Specifically, Supp. Fig. 7a presents all six individual marker channels from the iCLAP-IF staining, and Supp. Fig. 7b shows magnified panels. These zoomed-in images highlight distinct acinar cells that express either P21 (indicated by arrowheads) or P53 (indicated by asterisks), but not both, thereby supporting the conclusion that P21+ and P53+ cells represent separate cell populations in the acinar region.

The caption for this supplementary figure is also included below.

“Supplementary Figure 7: Reproducibility of staining pattern, UMAP embedding, and cluster occurrence across adjacent sections. a. Single-channel images of six senescence markers detected by iCLAP-IF, corresponding to Figure 3c. b. High-magnification images from the boxed acinar region in panel (a), showing single-channel P53 (red) and P21 (yellow) expression. These images show individual nuclei that distinctly express either P53 (asterisks) or P21 (arrowheads), but not both.”

6) "Next, we showed that the iCLAP workflow could be seamlessly integrated with conventional multiplex immunostaining techniques, including CyCIF, IMC, and CODEX, enabling 40+ plex detections in archival FFPE tissues (Fig. 4), while achieving high sensitivity detection on low abundance proteins." Only 12 markers are shown. IF 40+ markers have not been tested, revise statement.

We appreciate the reviewer's helpful feedback to clarify the extent of multiplexing demonstrated in our study. Because of the limited space, only a subset of markers is shown in the main figures for illustration, we have indeed tested higher-plex panels across multiple platforms. Complete antibody lists, clones, vendors, and dilutions are provided in Supp. Table 2. Specifically,

- The **iCLAP+CyCIF** panel consisted of 6 iCLAP and 6 CyCIF markers (12-plex total, all 12 plex were shown in **Fig. 4a**).
- The **iCLAP+CODEX** panel included 3 iCLAP markers and 28 CODEX markers (31-plex total, 12 plex were shown in **Fig. 5a**, and another 5 plex were shown in Supp. Fig. 18a; the full list could be found in Supp. Table 2).

- The **iCLAP+IMC** panel consisted of 3 iCLAP markers added to a 40-plex IMC panel (43-plex total, 12 plex were shown in **Fig. 5d**, and detailed intensity profile analysis for all 40 IMC panel markers were performed in **Supp. Fig. 19**).

To clarify the level of multiplexing achieved, we have revised the figure caption in Figure 5 and clarified the staining panel composition:

“Figure 5. Analysis of senescence and structural markers in pancreatic acini using multiplexed iCLAP-CODEX and iCLAP-IMC. a. Tissue imaging workflow using a 31-plex iCLAP-CODEX panel, composed of 3-plex iCLAP (P21, P53, P16) combined with a 28-plex CODEX antibody panel. Markers indicated with an asterisk (*) originate from iCLAP staining. Images display staining of P21*, P53*, P16*, panCK, E-cadherin, Ki67, CD45, α-SMA, and EPCAM in tissue sections, alongside an adjacent H&E-stained image. b. Representative images illustrating that P21+ cells are typically panCK+, whereas P53+ cells are generally panCK-. c. Quantitative analysis showing that 10% of P53+ cells are panCK+, while 65% of P21+ cells are panCK+, indicating distinct epithelial associations of these senescence markers. d. Tissue imaging using a 43-plex iCLAP-IMC panel, comprising 3-plex iCLAP (P21, P53, P16) combined with a 40-plex IMC panel. Representative images highlight colocalization patterns across different markers. The staining results also highlight the spatial association of P21 and PanCK.”

While we show representative 12-plex images from iCLAP+CODEX and iCLAP+IMC in the main figures, we provide full validation of the entire 40-plex iCLAP+IMC dataset in **Supp. Fig. 19** (reproduced below), which includes marker-by-marker signal intensity comparisons between conventional IMC and iCLAP-IMC staining.

Supp. Fig. 19

“Supplementary Figure 19. Comparison of iCLAP – IMC and conventional IMC staining across a 40 plex IMC panel. a-c. Representative images and signal intensity distributions for Collagen (a), Ki67 (b), and CD3 (c) comparing conventional IMC (solid lines) and iCLAP-IMC (dashed lines). iCLAP-IMC preserves comparable signal intensity with conventional IMC across distinct cellular and extracellular compartments, including the extracellular matrix (Collagen), nucleus (Ki67), and cell membrane (CD3). d. Signal intensity distribution plots for 36 additional markers spanning immune, stromal, and epithelial compartments. Across these markers, iCLAP-IMC demonstrates comparable signal intensity and distribution to conventional IMC, indicating effective preservation of marker signal across a broad multiplex panel. e. CD33 intensity distribution showing a reduced signal in iCLAP-IMC compared to conventional IMC. f–g. Despite the decreased CD33 intensity, colocalization with CD68-positive macrophages is retained, supporting the specificity and interpretability of CD33 staining in iCLAP-IMC.”

7) There is no mention of figure 4d or Supp. Fig. 10e-f in the text.

We thank the reviewer for catching this omission. We have now added explicit references to both **Fig. 4d** and **Supp. Fig. 10e–f** (now **Supp. Fig. 13a-f**) in the Results section to provide appropriate context and interpretation. The revised text in the manuscript now reads:

“Additionally, we observed a strong correlation between P16 intensity, insulin/glucagon intensity, and islet size, with larger islets exhibiting higher P16 intensity signal, higher glucagon intensity signal, and lower insulin intensity signal (Fig. 4d-e and Supp. Fig. 12c-e).

We found that as the islet sectional area increased, there was a corresponding rise in the incidence of P16+ and GCG+ cells, while the incidence of 53BP1+ and Insulin+ cells decreased (Supp. Fig. 13a-f).

Reviewer #2 (Remarks to the Author):

The focus of the manuscript of Wu and colleagues is to include the detection of low abundance proteins in established protocols using multiplex techniques for protein detection. To achieve this the three multiplex approaches CyCIF, CODEX, and IMC are used and combined with a tyramide signal amplification and bleaching workflow termed iCLAP. The novelty, the authors

report, is to improve the bleaching procedure to get several rounds of tyramide signal amplification. In combination with the multiplex approaches it is claimed that a cellular landscape of proteins can be visualized in combination with barely expressed proteins.

We appreciate the reviewer's clear summary of our goals and framing of the novelty. Our intent is precisely to enable repeated TSA-based detection within multiplex imaging by introducing an efficient bleaching/inactivation step, thereby allowing low-abundance proteins to be measured alongside high-plex panels (CyCIF, CODEX, IMC). We are grateful for the reviewer's positive assessment and thoughtful feedback for our work.

The approach is technically interesting as it extends the number of proteins to be analyzed in tissue samples. However, many issues remain unclear. Among various methodological points the most important one is that the authors want to investigate cellular senescence, especially in pancreatic tissue. A lot of data is collected, but in the end, it is not clear which in-depth analysis of cellular senescence the authors were able to reveal by their approach. Therefore, the manuscript needs to be substantially revised and improved.

Major comments:

Major comment 1:

The reader should get a better overview about the material used in this study. As senescence markers are in the focus of this work, and as age might influence the quality and structure of the tissue slices, it is important to get more information about the data shown. In the methods the authors mention that grossly normal pancreatic tissue of pancreatectomy specimens was used. Figure 3 shows data of a sample derived from a 63-year old donor. In figure 2, 4, and 5, where senescence markers and many other parameters were investigated and compared, and in the supplementary figures, we do not get to know anything about the samples. This has to be changed throughout and discussed adequately.

We thank the reviewer for this suggestion, we have now created a new **Supp. Table 1** that details the metadata associated with each sample analyzed throughout the manuscript including Figure reference, sample ID, Donor age, gender, sections ...etc. The new **Supp. Table 1** is now included in Tissue and Specimens section in the Materials and Methods section. The new table enables readers to directly link each figure panel to the underlying donor sample information (age and gender).

Major comment 2:

The TSA-technique was used for p16, p21, p53, and 53BP1 in the data presented in Figure 3. The signal-to-background-ratio looks quite well, however, a signal is only visible for p16 and 53BP1. I

would have expected positive signals for p21 and Lamin B1 somewhere in the slice as well. Is the same patient donor for the data presented in Suppl. Fig. 5 (where p21/Lamin B-positive cells are visible)? In the left part of Fig. 3c there are pink dots in the acinar tissue. Is this corresponding to p53 (indexed in red)? If yes, please, change the colours. Why did the authors decide to use the TSA technique for p16? This is also easy to detect by classical IF and I would say p16 is not a low abundant protein in islet cells.

We appreciate the reviewer's careful assessment and insightful questions regarding **Fig. 3**. To address the concerns thoroughly, we have separated the points into four parts and provide detailed responses below.

1. Lamin B1 Signal and Updated Images

Lamin B1 signal was not visible in the previously provided merged images due to an omission in the channel stack. We have now regenerated the merged images with the Lamin B1 channel included and updated Fig 3c. (reproduced below)

In addition, we added a new supplementary figure (**Supp. Fig. 7a**) displaying all six single-channel images corresponding to the ROI shown in **Fig. 3c** to clarify the spatial distribution and confirm the presence of Lamin B1 signal. **Supp. Fig. 7b** shows magnified panels. These zoomed-in images highlight distinct acinar cells that express either **P21** (indicated by arrowheads) or **P53** (indicated by asterisks), but not both, thereby supporting the conclusion that P21+ and P53+ cells represent separate cell populations in the acinar region.

2. Donor Consistency Across Figures

Yes, the data presented in **Supp. Fig. 5** and **Fig. 3** originate from the same donor. To improve transparency and help readers interpret the biological context of each figure, we have created a new **Supp. Table 1**, which lists the sample ID, section ID, age, gender, and tissue source for all samples used in both the main and supplementary figures.

3. Pink Signal in Acinar Tissue

We apologize for the confusion here, the apparent “pink” appearance results from the overlay of the red (P53) and blue (DAPI) channels in certain regions, rather than representing an independent pseudo color. This was clarified in the new single-channel images provided in **Supp. Fig. 7**, which allows better distinction between the individual markers. We have attached the **Supp. Fig. 7a below**:

Supp. Fig. 7a

4. Justification for TSA Amplification of P16

In addition to the P16 immunofluorescence (IF) staining with prolonged incubation (12 hours) presented in the original manuscript, we also performed P16 IF staining using a standard incubation time of 40 minutes, which was not included in the original submission. Under the standard incubation condition, the P16 signal was barely distinguishable from background (right panel below). Prolonged incubation (12 hours) resulted in a modest increase in signal intensity (**Fig. 2c**, reproduced below) and produced weakly positive staining in the islet region. However, even with prolonged incubation, the IF signal remained substantially lower than that achieved with TSA-based staining, with approximately an order of magnitude lower signal-to-noise ratio (**Fig. 2f-g**). Therefore, we consider the P16 as broader-line low abundant given the low signal detection with prolonged antibody incubations. Because of the limited space, we did not include the p16 staining from conventional workflow in our manuscript. We now explicitly address this in a new column titled “Performance with IF” in **Supp. Table 2**, where each antibody’s IF performance is categorized as “strong,” “weak,” “not detectable,” or “not evaluated,” based on intensity histograms. Under this criterion, the p16 antibody is weak and considered a low-abundance target requiring TSA amplification.

Major comment 3:

Starting at Figure 3d the authors focus on a detailed analysis of the whole pancreatic parenchyma. It remains unclear, where the 70,000 individual cells are from (proportion of tissue covered by exocrine vs. that covered by endocrine cells) and which selection parameters have been used. Is it one area (the one marked in part g of Fig. 3) or are different areas with more than one islet analyzed? How many islet cells were included in the 70,000 individual cells?

We thank the reviewer for the helpful suggestion. The ~70,000 cells analyzed were obtained from ten non-overlapping ROIs (each 2000 × 2000 pixels, approximately 1.3 mm × 1.3 mm) distributed across the pancreatic parenchyma to include endocrine, exocrine, and stromal compartments. ROI selection was based solely on image quality: we excluded section edges to maximize usable area and omitted visibly blurred or low-quality regions. No marker information was used during ROI selection. We have now reported the total analyzed area (10 ROIs × ~1.3 mm × 1.3 mm) in the figure caption.

Across these ROIs, 24 islets comprising 1,494 islet cells were annotated and analyzed. This information is now included in the revised **Supplementary Fig. 8c** (reproduced below), which summarizes the number of analyzed cells in each tissue category (acini, islet, duct, and stroma). In response to Reviewer 3's suggestion, we consolidated the cell clusters from 20 to 9 in the heatmap and UMAP (Fig. 3e–f and Supp. Fig. 8c) for improved readability, and we have updated the figures and legends accordingly.

Major comment 4:

Certainly, reproducibility is of great interest for the message of the paper. Are these clusters described in Fig. 4 reliably identified in samples of different donors? Which of the clusters are age-dependent? Can anything be said about gender differences? As Dapi is included here, we can see a huge variation in the nuclear staining in the cells. I would have expected that the Dapi signal is stable throughout and I would interpret a lower Dapi signal as a generally lower quality of the staining (maybe because of the quality of the tissue slice or else). How do you explain this variation?

For the analysis of acinar cells with iCLAP-CODEX it is also of great interest, if the correlation p21+/ductal component and p53+/acinar cells can be reproduced in different samples and is (in)dependent of the age of the donor, gender etc..

Comparisons between samples of different donors addressing the issues mentioned above should be added.

To verify technical reproducibility within the same donor, we applied the full 6 plex iCLAP-IF pipeline on an adjacent section and the results are now included in **Supp. Fig. 7** (reproduced below). We found that the resulting UMAP shows similar cluster distributions (**Supp. Fig. 7e-f**) using identical preprocessing and clustering parameters. Cluster composition across slides is also highly concordant (Pearson $r=0.959$, two-tailed $p=4.3 \times 10^{-5}$; 95% CI 0.813–0.992, Jensen–Shannon divergence = 0.0073; Hellinger distance = 0.071). Together, these results support the reproducibility of iCLAP-IF workflow. It should be noted that, in response to Reviewer 3’s comments, we have consolidated the number of clusters from 20 to 9 in **Fig. 3e-f** (reproduced below), which differs from the previous version of the manuscript.

In addition, we further applied the same staining panel to independent donors (two young, two old) to explore the reproducibility and aging associated effect. The results are now included in **Supp. Fig. 9** (reproduced below). The results show that p16+/53BP1+ staining in islets is consistently observed in both young and both old donors. However, p53+ or p21+ cells in acinar compartments are only observed in the two old donors (**Supp. Fig. 9a**). Quantitatively, older

donors show significantly higher occurrence of P16+ cells within islet (Supp. Fig. 9b-c). Together, these results indicate the iCLAP workflow can detect the senescence marker expression difference associated with aging.

We appreciate the reviewer's concern about apparent DAPI variability. From our understanding, uniform DAPI intensity across pancreatic compartments is not expected:

differences in chromatin compaction¹ and DNA content² among endocrine, acinar, and stromal nuclei—together with sectioning geometry (partial vs. equatorial nuclear profiles)—produce genuine intensity differences even under identical staining and imaging conditions.

We attached an image of DAPI staining (reproduced below) where intensity is indicated by the color. The ductal epithelial cells generally exhibit a stronger DAPI intensity in contrast to acinar cells that have relatively lower DAPI signals indicating that DAPI intensity is tissue compartment associated. Heterogeneity of staining is also observed within the same tissue compartment (i.e. acinar vs duct) which could likely be the result of variation in geometric of sectioning. Therefore, the observed variation reflects differences in tissue architecture and nuclear density across compartments, not inconsistency in staining quality.

Major comment 5:

It is under general discussion if one can really conclude from the intensity of a fluorescent antibody to the level protein expression. Fluorescence signals are quite difficult to standardize to protein expression. I would recommend to be more careful regarding this issue and to change the wording throughout the manuscript to “fluorescence intensity levels” or something equivalent.

We thank the reviewer for the suggestion. We have replaced “protein level” or “expression level” with “fluorescence intensity” throughout the manuscript.

Major comment 6:

In the last part of the results with iCLAP-CyCIF the authors say that “Overall, our results suggest that the expression of senescence markers is closely linked to the secretion of islet functional factors.”. The manuscript contains no functional data. It is “just” fluorescence signals indicating the existence of protein expression. Positivity for p16 and 53BP1 is associated with higher intensities for insulin and glucagon fluorescence staining. This might indicate more insulin/glucagon within the cell (although I would be very cautious about this, as we have archived embedded tissue), but it says nothing about secretion.

We have revised the manuscript text based on the reviewers’ comments. The phrase “secretion of islet functional factors” has been replaced with “insulin and glucagon intensity” to more accurately reflect what was measured. Additionally, throughout the manuscript, we have replaced “expression level” with more appropriate terms such as “intensity level” or “relative expression” where applicable, as previously suggested by the reviewer. The revised paragraph in the manuscript now reads:

“Cellular senescence is known to be associated with changes in morphology^{43–48}. Here, we found that nucleus size increased with 53BP1 intensity level. Nucleus size of relative 53BP1+ beta and delta cells was 50-60% larger than relative 53BP- beta cells. In contrast, relative P16+ expression was not correlated with nuclear size (Fig. 4i). Overall, our results suggest that the expression of senescence markers is closely linked to the insulin and glucagon intensity secretion of islet functional factors.”

Minor comments

(a) Suppl. Fig. 1: I assume two consecutive slices of the same sample were used for comparing the different methods, right? Please mention this in the figure legend. Please also comment on why the bleaching procedure has no influence at all on the blue nuclear Dapi staining.

Yes, the reviewer is correct that the CyCIF and iCLAP are adjacent (2 sections apart). In **Supp. Fig. 1**, CyCIF and iCLAP are performed in the adjacent sections for comparison. The main goal for the figures is to compare the bleaching efficiency between CyCIF and iCLAP. The reviewer is correct that both bleaching methods indeed significantly diminished the DAPI fluorescence signal. To ensure consistent visualization of tissue architecture and ROIs, we displayed the aligned first-round DAPI image as an overlay onto the post-bleaching images. We have further clarified that in the supplementary figure 1 captions. The figure caption now reads:

“Supplementary Figure 1. Comparison of iCLAP fluorophore removal methods vs CyCIF bleaching method. a. TSA-based high-intensity staining of CPA (red), GCG (yellow), and p16 (green) before and after one-hour iCLAP bleaching at various magnifications. **b.** TSA-based high-intensity staining of CPA (red), GCG (yellow), and p16 (green) before and after one or two hours of CyCIF bleaching at various magnification. Due to substantial DAPI signal reduction after iCLAP bleaching and CyCIF bleaching, the DAPI channel displayed represents the aligned DAPI signal from the first-round staining, used to maintain visualization of tissue architecture. The results demonstrate that the iCLAP bleaching method could effectively reduce TSA-based high-intensity staining to background levels, while the same duration or even longer periods of CyCIF bleaching still left noticeable residual fluorescence signals.”

(b) Suppl. Fig. 1b, right part: Please reformat the bars and sizes in the far right row. There are some overlaps.

We thank the reviewer for pointing this out, we have fixed this issue in **Supp. Fig. 1.**

(c) Suppl. Fig. 3: Please define the type of cells that was analysed in these images.

The analyzed cells in **Supp. Fig. 3** (now Supp. Fig. 4) correspond to acinar cells which were identified by the adjacent H&E staining. We have further clarified the cell types by including this information in figure captions.

(d) Suppl. Table 1: In the text it is said that 29 antibodies were tested and Suppl. Table 1 is given as a reference. I assume the sentence refers to those antibodies where it is indicated that TSA was used, but this are only 26. Please comment.

We thank the reviewer for pointing out this inconsistency. Initially, we excluded antibodies that were tested but not shown in figures, which led to the discrepancy in the reported number. We have now corrected this by including all 29 antibodies in the updated **Supp. Table 2.**

(e) What does the index “1” for Dapi mean in Fig. 4c?

We thank the reviewer for pointing out this confusion. The value "1" in the scale bar represents the Pearson correlation coefficient (ρ), which ranges from -1 to $+1$ and indicates the strength and

direction of linear correlation between pairs of islet-level features. The diagonal values are equal to 1, as each feature is perfectly correlated with itself. We updated the **Fig. 4** caption to clarify this.

(f) Statistical analysis: I assume that the analysis describes the data presented in Fig. 1d. Please add this information, and also add if any other statistical evaluation was performed.

We appreciate the reviewer's suggestion. We have revised the statistical analysis section to explicitly state that the analysis corresponds to **Fig. 1d**.

Reviewer #3 (Remarks to the Author):

The authors present integrable Co-detection of Low Abundant Proteins (iCLAP) which uses tyramide signal amplification (TSA) followed by chemical bleaching to integrate detection of lowly-expressed antigens via enzymatic deposition of tyramide molecules with highly-multiplex imaging techniques CyCIF, CODEX and IMC. The authors demonstrate complete fluorophore bleaching using their novel method, improved sensitivity over conventional IF, and integration with other multiplex methods. For example, integration with CyCIF enabled islet cell typing with insulin, glucagon and somatostatin 28 and revealed a negative correlation with islet size/P16 expression and insulin expression.

The innovation in this work is the novel fluorescence bleaching method which is 2M H₂O₂ (almost 7%) and 3mM EDTA in PBS at pH 12.5. Lin et al. previously published a bleaching method using 4.5% H₂O₂ and 24 mM NaOH made up in PBS, and others have demonstrated 20 mM NaOH and 3% H₂O₂ in PBS, pH 7.4 [PMID: 35545666]. Both works demonstrated 70-80% tissue loss over 10 rounds using these conditions. The authors should quantify tissue loss in the iCLAP protocol, especially in normal tissues which tend to be more fragile in the two above-mentioned references. Tissue loss could be especially likely with the additional rounds of antigen retrieval inherent in the TSA protocol.

Pulsawatdi et al. demonstrated that antigenicity changes after multiple rounds of HIERS, and Eng et al. showed that repeated rounds of H₂O₂ quenching and blocking also alter non-specific background and dynamic range. Quantification of these metrics for iCLAP versus single plex would be informative. For example, in supplemental figure 5, LaminB1 staining looks dimmer in iCLAP-IF than in single plex but there is no quantification. Finally, reproducibility is an open question. While it had been addressed in each technology individually [PMID: 34266881, 35545666] I believe the authors should quantify the reproducibility of combining the TSA and

(CyC)IF methods into iCLAP.

Overall, the authors report a useful method to combine well-established TSA protocols and multiplex imaging methods (IF, CyCIF, CODEX). As they state, iCLAP could be a “powerful tool for studying various critical biological processes in situ” however, additional quantitative metrics for tissue loss, SNR compared with single plex, and reproducibility would be informative for researchers wishing to adopt this method.

We thank reviewer for this suggestion and appreciate the reviewer’s positive assessment. The requested metrics—tissue retention, SNR/dynamic range vs single-plex (including Lamin B1), and reproducibility, references—have been addressed in detail in the specific comment responses, with corresponding updates to the manuscript. We believe these additions substantially improve the re-usability of the iCLAP protocol.

Specific comments:

1. Figure 1: Quantify any tissue loss due to bleaching and antigen retrieval. This should be done on the tissue microarrays containing normal breast and skin for best evaluation.

We thank the reviewer for the suggestion. We have performed a systematic comparison of tissue integrity across 11 rounds of staining in both the iCLAP and CyCIF workflows, using normal breast and pancreas tissues. The results are now summarized in a new **Supplementary Fig 2** (reproduced below). In the iCLAP workflow, each staining round begins with three sequential antibody elution steps (we are using chemical elution instead of heat antigen retrieval steps to strip primary antibodies), followed by counterstaining and iCLAP bleaching (**Supp. Fig 2a**). Imaging is performed after every two staining-bleaching cycles (i.e., 1st, 3rd, 5th, etc.), up to the 11th round. In contrast, the CyCIF workflow (bottom) involves direct counterstaining and CyCIF bleaching steps without antibody elution steps, with imaging also conducted at every odd-numbered round. Whole-slide images of DAPI-stained nuclei of normal breast and pancreas tissues readily illustrate minimal tissue loss between the 1st and 11th rounds (**Supp. Fig. 2d-e**). We further assess the tissue integrity by nuclei counts in round 1 and round 11 in 10 ROIs (2000 × 2000 px; ~1.3 × 1.3 mm each). Our results show that on average, >95% of nuclei remain after round 11 in both iCLAP and CyCIF workflow (Supp. Fig. 2b-c) in both tissue types. We noted nuclei loss if slightly less in pancreas tissue than breast tissue as expected. High-magnification images from normal breast stromal regions and pancreatic parenchyma further confirm consistent cell density and indicate that tissue retention remains comparable between CyCIF and iCLAP workflows throughout all staining rounds. Together, these analyses demonstrate that the iCLAP workflow does not compromise tissue integrity relative to the CyCIF workflow, and that both methods maintain stable tissue retention (>95%) over extended staining cycles.

Supp. Fig.2

“Supplementary Figure 2. Comparison of tissue retention across multiple rounds of staining using iCLAP and CyCIF workflows. a. Schematic overview of the tissue processing workflows in iCLAP and CyCIF across 11 staining cycles. In the iCLAP workflow (top), each staining round begins with three sequential antibody elution steps, followed by counterstaining and iCLAP bleaching. Imaging is performed after every two staining-bleaching cycles (i.e., 1st, 3rd, 5th, etc.), up to the 11th round. In contrast, the CyCIF workflow (bottom) involves direct counterstaining and CyCIF bleaching steps without antibody elution steps, with imaging also conducted at every odd-numbered round. **b-c.** Quantification of tissue retention across staining rounds in normal breast (**b**) and pancreas (**c**) tissues. Tissue area was measured separately in epithelial and stromal regions across 11 rounds, showing similar retention between CyCIF workflow and iCLAP workflow. **d-e.** Whole slide images of normal breast (**d**) and pancreas (**e**) tissues showing preservation of DAPI-stained nuclei from round 1 (green) and round 11 (red), with minimal tissue dropout in both iCLAP workflow. **f-g.** High-magnification images from normal breast stromal regions (**f**) and pancreatic parenchyma (**g**) confirm consistent nuclear density and indicate that tissue retention remains comparable between CyCIF and iCLAP workflows throughout all staining rounds.”

Notably, the iCLAP bleaching solution did not produce bubbles during the entire bleaching process, whereas the CyCIF bleaching solution continuously generated bubbles (see unpublished figure below). This difference may account for the comparable degree of tissue loss observed between the CyCIF and iCLAP bleaching methods.

2. Figure 2: Quantify SNR of conventional IF p16.

We thank the reviewer for this helpful suggestion. We would like to clarify that SNR quantification for conventional IF p16 is already included in **Fig. 2f**. To ensure this is more clearly visible, we have bolded and boxed the SNR values in the updated figure panels (2f and 2g):

3. Supplemental 5: quantify signal to noise ratio of single plex and iCLAP-IF.

We thank the reviewer for the suggestion. We have further evaluated the staining signal comparing single-plex vs iCLAP-IF. In general, due to the lack of a universal “positive” cell population for many markers, we assessed signal-to-noise using intensity histograms, rather than relying on predefined thresholds for positivity. We plotted intensity histograms of single-plex TSA or IF and iCLAP-IF for all six markers. These histograms showed intensity histogram profile are quantitative consistent between single-plex and iCLAP-IF. These results are further validated by representative images using identical intensity thresholds that demonstrate comparable staining intensity and spatial patterns across both methods. These results confirm that iCLAP-IF maintains signal intensity and distribution comparable to single-plex TSA or IF staining. We have included these data as **Supp. Fig. 6** reproduction below.

Supp. Fig. 6

“Supplementary Figure 6. Multiplex iCLAP-IF preserves signal intensity compared to single-plex staining. a–f. Representative images (top) show staining for p16 (a), 53BP1 (b), p21 (c), p53 (d), HMGB1 (e), and Lamin B1 (f) using conventional single-plex TSA/IF and multiplex iCLAP-IF. Quantitative fluorescence intensities (bottom, log₁₀ scale) show that iCLAP-IF maintains distributions comparable to single-plex staining, confirming minimal loss of sensitivity with six-plex detection.”

4. Figure 3: Quantify the reproducibility of the iCLAP-IF method across adjacent slides.

We applied the full 6 plex iCLAP-IF pipeline on an adjacent section and the results are now included in **Supp. Fig. 7** (reproduced below). We found that the resulting UMAP shows similar cluster distributions (**Supp. Fig. 7e-f**) using identical preprocessing and clustering parameters. Cluster composition across slides is also highly concordant (Pearson $r=0.959$, two-tailed $p=4.3 \times 10^{-5}$; 95% CI 0.813–0.992, Jensen–Shannon divergence = 0.0073; Hellinger distance = 0.071). Together, all these results support the reproducibility of iCLAP-IF.

Supp. Fig. 7e-f

Also, clustering the heatmap in figure 3e would improve readability, especially with a dendrogram to show the similarity of senescence marker clusters as shown on the umap in f.

We thank the reviewers for this helpful suggestion. We have consolidated the 20 clusters into 9 by manually merging those with similar expression patterns and added a dendrogram in **Fig. 3e** based on hierarchical clustering. The figure and legend have been updated accordingly

Fig. 3e

5. Supplemental 7: how much, if any tissue loss was experienced in these TMAs? Normal breast and normal skin have shown particular fragility in other platforms.

We thank the reviewer for raising this important point. We have already conducted a more comprehensive, well-controlled analysis of tissue retention using whole-slide sections of normal breast and pancreas tissues as suggested (Specific comment 1 and **Supp. Fig. 2**). That dedicated experiment demonstrated >95% retention across 11 cycles, with no significant difference between iCLAP and CyCIF workflows, thereby directly addressing concerns about tissue fragility.

The specific goal of Supp. Fig. 7 (now **Supp. Fig. 10**) was to demonstrate that the iCLAP 6-plex panel enables robust detection of senescence markers across diverse FFPE tissue types, rather than to readdress tissue integrity.

6. In figure 4, please report the p-value as well as rho for the Pearson correlation.

We thank the reviewer for the suggestion. In the revised version of **Fig. 4**, we have now included the p-values alongside the Pearson correlation coefficients rho (ρ) for each panel.

7. Supplemental figure 9: add p-values on Pearson correlations, add Pearson correlation for f-g.

We have updated Supp. Figure 9 (now **Supp. Fig. 12**) to include both the Pearson correlation coefficients rho (ρ) and their corresponding p-values in all relevant panels.

8. Supplemental figure 10c is missing the x-axis label. 53BP1 representative images should be shown (quantified in b).

We thank the reviewer for pointing out these omissions. We have updated Supp. Fig. 10c (now **Supp. Fig. 13c**) to include the missing x-axis label.

We also included representative iCLAP-CyCIF images of 53BP1 staining in Supp. Fig. 10e-f (now **Supp. Fig. 13e-f**, reproduced below), illustrating that larger islets exhibit reduced occurrence of 53BP1-positive cells compared to smaller islets:

Supp. Fig. 13e-f

9. Supplemental figure 12 shows no quantification to support the lack of association between P21, P53 and acinar enzyme expression. Unsupervised clustering on single cell mean intensity would support this claim.

We thank the reviewer for this helpful suggestion. In response, we have added quantitative analyses to support our claim that P21 and P53 signals are not associated with acinar enzyme expression. These new analyses are included in the updated and expanded Supplementary Figure 12 (now **Supplementary Figure 15**). Specifically, we visually identified regions with high or low P53+ cell occurrence (five ROIs each) and regions with high or low P21+ cell occurrence (five ROIs each) (**Supp. Fig. 15a–d**). Quantitative analysis confirmed that the selected regions indeed exhibited distinct P53+ or P21+ cell occurrences (**Supp. Fig. 15e, f**). We then measured the mean staining intensity of CPA, lipase, and trypsin (digestive enzymes) within these ROIs and found no statistically significant differences between regions with high versus low P53+ or P21+ cell densities (**Supp. Fig. 15g, h**). Together, these new results further support the lack of association between P21/P53 status and acinar enzyme expression.

Supp. Fig. 15

“Supplementary Figure 15: Lack of association between P21+/P53+ cell density and acinar enzyme expression. a-d. Representative iCLAP-CyCIF images co-detecting P53 or P21 with acinar enzyme markers CPA, Trypsin, and Lipase in the P53-high (a), P53-low (b), P21-high (c), and P21-low (d) acinar regions. Adjacent H&E staining provides tissue morphology for reference. No visual co-variation between P53/P21 occurrence and enzyme signal is evident. e-f. Quantification of P53+ (e) and P21+ (f) cell density confirms that the designated high vs low regions differ as intended. g-h. Mean enzyme intensities across five ROIs per group show no statistically significant differences between P53-high vs P53-low (g) or P21-high vs P21-low (h) regions. Statistical tests and *P* values are described in the Statistical Analysis section.”

10. The authors state regarding Figure 5 c that “P53+ cells arise from acinar cells”. I don’t agree that lack of panCK in P53+ cells means they have a different cell of origin. This statement and the section title "distinct cellular origins of P21+ and P53+ cells in the acinar region” should be replaced with a less strong claim.

We thank reviewer for this suggestion. We have changed the section title to: **iCLAP-CODEX and iCLAP-IMC reveal distinct epithelial marker expression of P21+ and P53+ cells in the acinar region.** The revised main text is reproduced below:

“Next, we evaluated the marker profiles associated with P21+ and P53+ cells in the acinar compartment. The iCLAP-CyCIF analysis showed no significant differences in the intensity patterns of acinar enzymes (CPA, trypsin, and lipase) within regions exhibiting a high density of P21+ or P53+ cells (Supp. Fig. 15a–h). Using iCLAP-CODEX, we simultaneously detected three senescence markers (P21, P53, and P16) alongside a 28-plex CODEX panel of tissue and cell markers, including panCK, E-cadherin, EpCAM, Ki67, α -SMA, collagen, CD3, CD45, and CD68 (Fig. 5a).”

11. Figure 5 d lacks validation of IMC staining compared to standard IMC. For example, while Vim and collagen seem to match the H&E, the staining seems weak and diffuse. The iCLAP + IMC data should be quantitatively compared to standard IMC.

We thank the reviewer for raising this important point regarding validation of IMC staining in the iCLAP-IMC context. To directly address this, we have performed conventional IMC staining and imaging without iCLAP in the same sample for comparison. We have performed quantitative comparison between standard IMC and iCLAP-IMC staining across a 40-plex panel (shown in

Supp. Fig. 19, reproduced below). For each marker, we plotted the signal intensity distributions from both IMC and iCLAP-IMC (solid vs. dashed lines, respectively). The results demonstrate that majority of markers (39 out of 40) have closely matched signal intensity profiles, including both Vimentin and Collagen. We found only CD33 staining by IMC after iCLAP shows significantly drop in signals compared to IMC direct. Though the low intensity signal of CD33, we found the CD33 positive cells detected in iCLAP-IMC are colocalized with CD68 as expected and demonstrate the CD33 detection still valid in amid the reduced intensity.

“Supplementary Figure 19. Comparison of iCLAP – IMC and conventional IMC staining across a 40 plex IMC panel. a-c. Representative images and signal intensity distributions for Collagen (a), Ki67 (b), and CD3 (c) comparing conventional IMC (solid lines) and iCLAP-IMC (dashed lines). iCLAP-IMC preserves comparable signal intensity with conventional IMC across distinct cellular and extracellular compartments, including the extracellular matrix (Collagen), nucleus (Ki67), and cell membrane (CD3). d. Signal intensity distribution plots for 36 additional markers spanning immune, stromal, and epithelial compartments. Across these markers, iCLAP-IMC demonstrates comparable signal intensity and distribution to conventional IMC, indicating effective preservation of marker signal across a broad multiplex panel. e. CD33 intensity distribution showing a reduced signal in iCLAP-IMC compared to conventional IMC. f–g. Despite the decreased CD33 intensity, colocalization with CD68-positive macrophages is retained, supporting the specificity and interpretability of CD33 staining in iCLAP-IMC.”

Supp. Fig. 19

12. Supplemental figure 13 would benefit from quantification showing P53+ and P21+ cells are immune marker negative.

We thank the reviewer for this insightful suggestion. We have performed the quantitative analyses of occurrence of co-expressed CD45 with p53 or p21 and expanded **Supp. Fig. 13** (now **Supp. Fig. 16**, reproduced below) with this analyses results. We found that CD45+ cells are generally neither P53+ or P21+ where only 3 cells out of 374 CD45+ cells are either P53+ or P21+ (**Supp. Fig. 16b-c**). We also quantified the occurrence of CD45+ and CD45- populations within P53+ and P21+ cells. The bar graph in **Supp. Fig. 16d** shows the occurrence of CD45+ versus CD45- staining specifically within P53+ and P21+ populations. This demonstrates that both P53+ and P21+ cells were significantly enriched in CD45- populations, confirming their non-immune status.

These findings are consistent with panel a, further supporting that P53+/P21+ cells in acinar is not associated with immune lymphocytes.

Supp. Fig. 16

“Supplementary Figure 16. P21+ and P53+ acinar cells are distinct from immune cell populations. a. Representative multiplex iCLAP-IF images of human pancreatic tissue sections stained for P21 (yellow) and P53 (red) with immune markers CD3, CD8, CD14, CD40, CD45, CD56, CD68, and CD141 (green) in the acinar region. Arrows indicate P21+ cells, asterisks indicate P53+ cells, and arrowheads indicate immune marker-positive cells. Lack of immune marker expression confirms that P21+ and P53+ cells are non-immune. Adjacent H&E staining is shown for morphology reference. b-c. Venn diagrams show minimal overlap between P53+ or P21+ populations and CD45+ immune cells in 3 immune-enriched ROIs (n = 2322 total cells). d. Quantification from 3 immune-enriched ROIs (dots = per ROI values; bars = mean \pm SD) confirms that P53+ and P21+ cells are enriched in CD45- populations. Statistical tests and P values are described in the Statistical Analysis section.”

In summary, beautiful staining of low abundance targets. As a potential user, I wonder if the authors could comment on the difficulty of optimizing panel order. Are there any lessons learned during the development of this method?

We thank the reviewer for the encouraging feedback and their constructive suggestions regarding technological development and here are some lessons we learned during the development of iCLAP:

1. Panel order design (intensity-tiering and sequence).

We begin with a uniform test protocol (same primary concentration, fluorophore, incubation) to obtain per-target intensity distributions on representative tissues. We then rank targets from low to high apparent abundance and set the TSA staining sequence accordingly:

- High-abundance targets (that are detectable by conventional IF with good signal) are placed later in the sequence and will be stained with normal IF. We stained multiple targets using conventional IF and TSA, which provided a benchmark for the signal levels that produce good IF readouts with our protocol.
- Low abundance/weak targets were detected with TSA in sequence based on their relative intensity with the uniform testing protocol. The TSA targets were then titrated so that peak intensities are comparable across rounds. This minimizes channel imbalance, reduces apparent “leak,” and decreases the primary residue across rounds.

- Finally, we validated the multi-round panel by comparing single-plex and iCLAP images for intensity concordance and morphological consistency. We also performed round-swap staining of TSA targets on adjacent sections to verify that serial TSA introduces negligible masking/steric-hindrance effects.

2. Antibody selection and validation

We invested substantial effort and resources in antibody validation. For example, we screened >10 p16 antibodies from multiple vendors; some produced predominantly nuclear staining, whereas most yielded cytoplasmic patterns. Selection was guided by literature and clinical context, together with extensive positive/negative controls. We prioritized antibodies for which independent clones/vendors showed concordant subcellular localization and morphology.

3. In terms of the antigenicity

Across three years of iterative testing using two archival pancreas specimens, antigenicity dropped substantially after ~1 year when slides were kept at room temperature. We therefore store sections at 4 °C with desiccant, which largely preserves antigenicity relative to room-temperature storage.

REVIEWER COMMENTS

Reviewer #1 (Remarks to the Author):

Thank you to the authors for addressing my comments. I have no further remarks.

We would like to thank Reviewer #1 for their thoughtful feedback throughout the revision process. Their insightful comments had greatly improved the overall integrity and consistency of our manuscript.

Reviewer #2 (Remarks to the Author):

In the revised manuscript of Wu and colleagues my issues are addressed adequately. During re-evaluation of my former major comment 6, I recognized that the text box marked as revised text does not fully match with the text in the revised manuscript (lines 209 to 295). I assume something is wrong with the copied part in the answer-to-referees-letter. As the altered text in the manuscript is o.k., it doesn't matter, I was just wondering, what happened.

Two minor points that should be changed:

1. I recommend to add scale bars to all panels shown in the Supplementary Data (missing e.g. in the new version of Suppl. Fig. 9 and in Suppl. Fig. 15).
2. Fig. 3c and new Suppl. Fig. 7a and b: Please change „um“ to “ μm ”.

Thank you for your careful inspection of our manuscript and for pointing out this discrepancy. We believe the comment refers to lines 290–295 of the revised manuscript, rather than lines 209–295 as stated in our response letter. During the revision process, multiple rounds of edits were made to both the manuscript and the response-to-reviewers document. We apologized that the response letter was not fully updated to reflect this change. The text in the revised manuscript represents the final and intended version. We apologize for the confusion.

Following your suggestions, we have added scale bars to all panels in the Supplementary Data, including Supplementary Figures 9 and 15, to ensure clarity and consistency and we have corrected the unit "um" to " μm " in Figure 3c and Supplementary Figures 7a and 7b.

We appreciate the reviewer's careful checking to details.

Reviewer #3 (Remarks to the Author):

The authors have satisfactorily addressed all of my previous concerns, including tissue retention, signal to noise ratio, reproducibility and statistics and the manuscript is now suitable for publication.

We sincerely thank Reviewer #3 for the constructive feedback throughout this process from a technical perspective, which has significantly enhanced the quality and impact of our work.